# Coronary Artery Spasm-Related Heart Failure Syndrome: Literature Review

**DOI:** 10.3390/ijms24087530

**Published:** 2023-04-19

**Authors:** Ming-Jui Hung, Chi-Tai Yeh, Nicholas G. Kounis, Ioanna Koniari, Patrick Hu, Ming-Yow Hung

**Affiliations:** 1Division of Cardiology, Department of Internal Medicine, Chang Gung Memorial Hospital Keelung, Chang Gung University College of Medicine, Keelung City 24201, Taiwan; hmj1447@cgmh.org.tw; 2Department of Medical Research and Education, Shuang Ho Hospital, Taipei Medical University, New Taipei City 23561, Taiwan; ctyeh@s.tmu.edu.tw; 3Continuing Education Program of Food Biotechnology Applications, College of Science and Engineering, National Taitung University, Taitung 95092, Taiwan; 4Department of Cardiology, University of Patras Medical School, 26221 Patras, Greece; ngkounis@otenet.gr; 5Cardiology Department, Liverpool Heart and Chest Hospital, Liverpool L14 3PE, UK; ioanna.koniari@lhch.nhs.uk; 6Department of Internal Medicine, School of Medicine, University of California, Riverside, Riverside, CA 92521, USA; patrick.hu@ucr.edu; 7Department of Cardiology, Riverside Medical Clinic, Riverside, CA 92506, USA; 8Division of Cardiology, Department of Internal Medicine, Shuang Ho Hospital, Taipei Medical University, No.291, Zhongzheng Rd., Zhonghe District, New Taipei City 23561, Taiwan; 9Taipei Heart Institute, Taipei Medical University, Taipei City 110301, Taiwan; 10Division of Cardiology, Department of Internal Medicine, School of Medicine, College of Medicine, Taipei Medical University, New Taipei City 23561, Taiwan

**Keywords:** heart failure with preserved ejection fraction, heart failure with reduced ejection fraction, coronary artery spasm

## Abstract

Although heart failure (HF) is a clinical syndrome that becomes worse over time, certain cases can be reversed with appropriate treatments. While coronary artery spasm (CAS) is still underappreciated and may be misdiagnosed, ischemia due to coronary artery disease and CAS is becoming the single most frequent cause of HF worldwide. CAS could lead to syncope, HF, arrhythmias, and myocardial ischemic syndromes such as asymptomatic ischemia, rest and/or effort angina, myocardial infarction, and sudden death. Albeit the clinical significance of asymptomatic CAS has been undervalued, affected individuals compared with those with classic Heberden’s angina pectoris are at higher risk of syncope, life-threatening arrhythmias, and sudden death. As a result, a prompt diagnosis implements appropriate treatment strategies, which have significant life-changing consequences to prevent CAS-related complications, such as HF. Although an accurate diagnosis depends mainly on coronary angiography and provocative testing, clinical characteristics may help decision-making. Because the majority of CAS-related HF (CASHF) patients present with less severe phenotypes than overt HF, it underscores the importance of understanding risk factors correlated with CAS to prevent the future burden of HF. This narrative literature review summarises and discusses separately the epidemiology, clinical features, pathophysiology, and management of patients with CASHF.

## 1. Introduction

Heart failure (HF) is a clinical, heterogeneous syndrome stemming from any structural or functional ventricular impairment of diastolic filling or systolic ejection fraction or both [1]. Because the left ventricular ejection fraction (LVEF) has a bimodal distribution among HF patients [2], LVEF has been a phenotypic marker indicative of idiosyncratic pathophysiological mechanisms [3,4] and, most importantly, response to therapies [5]; however, regardless of LVEF, the prognosis of HF patients has been also correlated with diastolic dysfunction [6]. In the general population, LV diastolic dysfunction demonstrates a prevalence of ~21%, but only 1.1–5.5% of individuals present symptoms [7], suggesting the ischemic cascade beginning with clinically silent diastolic dysfunction has been substantially underrecognized. There is no specific, noninvasive diagnostic test that serves as a gold standard for HF diagnosis since it is a clinical entity established upon a careful history, physical examination, laboratory, and imaging data. The clinical diagnostic gold standard of HF is the identification of an elevated pulmonary capillary wedge pressure at rest or exercise on an invasive hemodynamic exercise test in a symptomatic patient [8]. While most patients with suspected HF do not require invasive testing for diagnosis, an echocardiogram constitutes often the best method for HF diagnosis [9].

In the US, approximately 50% of HF patients have LVEF ≥50%, with the balance having LVEF <50% [10,11]. While HF with reduced ejection fraction (HFrEF) <50% is a final common pathway of systolic dysfunction due to various etiologies [12], numerous drugs and cardiac devices have been reported to enhance outcomes in patients with HFrEF independently of etiology, demonstrating that patients with HFrEF share similar pathophysiological pathways to the progression of systolic dysfunction [12]. In contrast to HFrEF, such a unifying pathophysiological adaptation is lacking in HF with preserved ejection fraction (HFpEF) ≥50% [13], which has proved to be the main form of HF worldwide because of aging of the general population and the augmenting prevalences of obesity, diabetes mellitus, and hypertension [13]. Because the relevant therapy should target the different underlying etiologies, pathophysiologies, and comorbidities [3,14,15,16], patients with HFpEF may respond in a less homogenous way to treatment. Furthermore, the long-term survival rates in HFpEF patients are lower than those in HFrEF patients, although mostly driven by non-cardiovascular causes [17]. Notably, HF is considered as treatment failure rather than an indication for therapy [18]; future attempts to reduce HF burden should focus not only on reducing or averting exposure to risk factors but also on the management of comorbidities. However, neither the molecular mechanisms underlying HF, irrespective of LVEF, nor effective prevention strategies are fully understood.

Basic science evidence, epidemiological studies, and clinical trials suggest that coronary artery disease (CAD), including epicardial CAD and coronary microvascular disease (CMD) [19], is a significant contributor to HF pathogenesis. In patients with HFrEF, CAD constitutes frequently the main cause [20]. Thus, up to 25% of HF patients classified clinically as ‘‘nonischemic cardiomyopathy,’’ might reveal evidence of CAD at autopsy [21], and ischemic changes have also been reported in endomyocardial biopsies [22] in such patients. On the other hand, coronary artery spasm (CAS), an excessive coronary vasoconstriction leading to total or subtotal vascular obstruction, has been considered one of the causes of HFrEF [23,24,25].

As a separate entity from classic angina pectoris (pectoris dolor) described by Dr. William Heberden (1710–1801) based on 20 cases with this affliction in 1772 [26], which started when chest pain was evoked by exercise or emotional stress, and was relieved by rest or nitroglycerin, in 1959, with Dr. Rexford Kennamer and others, Dr. Myron Prinzmetal (1908–1987) published the first landmark report [27] of their findings on “A variant form of angina pectoris”. In Prinzmetal’s first report, among the 32 cases of variant angina between 1931–1956, of which 20 and 12 were personally observed and reported in the literature, respectively, the pain appeared at rest or during daily routine activity but was not caused by exercise or emotional stress. Among the reported 32 patients, 12 had myocardial infarction [MI] at follow-up [27]. Because both forms of angina pectoris had coronary atherosclerosis in common post-mortem, and the variant angina attack usually happened at rest, when the vascular physiologically hypertonic action is greatest [27], the mechanism for variant angina proposed by Prinzmetal et al., or other researchers was arterial hypertonus or CAS, respectively. Of note, following the first report of coronary angiography in 1959, CAS in variant angina had never been proved angiographically [28,29] within a decade until the early 1970s [29,30,31].

It was common for coronary angiography in the 1970s and 1980s to diagnose CAS in the catheterization laboratory. It turned out increasingly clear that CAS could occur in a patient with [26,27] or without atherosclerotic obstructive CAD, referred to as “variant of the variant” [30]. Moreover, CAS is more frequently associated with ST-segment depression rather than non-progressive elevation [32,33]. Hence, the term “variant angina” is specifically reserved for CAS-induced angina with temporary ST-segment elevation (Figure 1). In addition, CAS-related acute coronary syndrome can be due to anaphylactic reactions, involving the release of inflammatory mediators such as histamine, chymase, leukotrienes, and platelet-activating factor from mast cells upon activation to cause the constriction of coronary vascular smooth muscle cells that constitute the pathophysiologic mechanism of Kounis syndrome [34,35]. Altogether, in coronary heart disease, atherosclerotic obstructive CAD cannot be regarded as the sole source of angina pectoris [36].

Most importantly, the use of nitroglycerin at the beginning of coronary angiography should be avoided [37] to prevent inadvertent abrogation of spontaneous CAS. However, nitroglycerin solution has to be fully prepared before performing CAS provocative testing to relieve established CAS promptly through intracoronary infusion [33]. Therefore, 2 sets of coronary angiograms pre- and post-intracoronary nitroglycerin should be obtained routinely once obstructive lesions are noted. Spontaneous CAS can be misdiagnosed as a candidate for percutaneous coronary intervention unless the alleviation of obstructive stenosis is documented after intracoronary nitroglycerin, emphasizing the importance of intracoronary nitroglycerin infusion before endeavored coronary intervention, and avoiding unnecessary coronary revascularization [38]. Because coronary revascularization in selected obstructive CAD patients can ameliorate diastolic dysfunction, decrease morbidity and mortality [39,40,41,42], and enhance systolic function [43], likely by improvement of hibernating myocardium, medical treatments in non-obstructive CAD, such as CAS, patients may similarly improve left ventricular diastolic and systolic function.

## 2. Epidemiology

Framingham Heart Study has demonstrated that, from the 1950s onwards, the role of myocardial ischemia and infarction has developed substantially [44,45], placing great emphasis on the prevention of HF through the prevention of myocardial ischemia and infarction. Despite the fact that the cumulative incidence of HF is similar between both genders, women are approximately 65% less likely than men to develop HFrEF, particularly in their younger years [45,46,47], while HFpEF is twice as common in women than men, which results from physiologic differences between the two genders [48]. On the other hand, survival after a diagnosis of HF, irrespective of HFrEF or HFpEF, has shown modest improvement in the 21st century and lags behind other serious conditions, such as cancer [49]. Hence, contributing factors require further clarification, among which CAS is becoming important. There are large variations in CAS prevalence across the world, as CAS frequency is greater in Japan than in west countries [50]. The prevalence of CAS is high (40%) among patients showing evanescent, resolving ST-segment elevation admitted to Japanese hospitals [51]. Moreover, using provocative testing, CAS of more than two coronary arteries appears more frequently in Japanese (24.3%) [52] and Taiwanese populations (19.3%) [53] than in Caucasians (7.5%) [54]. On the other hand, men are more likely than women to develop CAS both in East Asia and Western countries [51,53]. Most CAS appears in people aged 40 to 70 years and the prevalence declines after the age of 70 years [27,51,53]. Several studies have demonstrated that CAS prevalence in patients without obstructive CAD is around 50% in angina and, specifically, 57% in acute coronary syndrome in Asia [55,56,57]. Among provocative tests using intracoronary acetylcholine for functional vasomotor abnormalities in acute coronary syndrome without obstructive CAD, 79% of individuals demonstrate a positive finding in Japan [58], whereas the results are positive in 16% of French [59] and 49% of German [57] patients. Notably, CAS diagnosis can be challenging due to pretreatment with antispastic nitroglycerin or calcium channel antagonists, refraining from coronary constrictors, and changes in disease activity. Additionally, contemporary trends of CAS prevalence tend to decline in Japan because of reduced performance of misperceived time-consuming provocative tests, or extensive use of statins and calcium channel antagonists [58].

CAS is an exceptionally complex multifactorial disease in which smoking, inflammation, metabolic, psychosocial, and physical factors come into play. Although it was reported more than 20 years ago [50], the racial differences in coronary vasomotion disorders between Asian and Caucasian populations remain controversial. First, previous studies show that epicardial CAS is more often recognized in Japanese and Taiwanese people than in Caucasian populations, while CMD is typically observed in Caucasian patients, which may be because Japanese and Taiwanese cardiologists have performed spasm provocation testing actively for 30 and 20 years, respectively, in patients with nonobstructive CAD, whereas most Caucasian cardiologists do not perform provocative testing for nonobstructive CAD in the cardiac catheterization laboratory [60]. However, for an unknown reason, some Taiwanese cardiologists are resistant to acknowledging the existence of CAS, which affects patients’ physical and psychological quality of life, and as a result, oppose performing provocative testing for the diagnosis of CAS. Second, various diagnostic procedures are performed worldwide, such as intravenous ergonovine-provoked >70% luminal reduction in France [61] and intracoronary acetylcholine-provoked >75% luminal reduction in Germany [57]. Third, according to a Japanese study [62], intracoronary acetylcholine administration time is crucial to provoke CAS. Slow injection of acetylcholine for 3 min may induce microvascular CAS, whereas rapid injection of acetylcholine for 20–30 s may provoke epicardial CAS, leading to inconsistency in the prevalence and incidence of CAS between Japanese and Caucasian patients. Fourth, Japanese cardiologists have stated that in some European institutions, acetylcholine testing without pacemakers is employed, which may cause bradycardia or cardiac arrest in the right coronary artery rendering difficult interpretation of provocative testing. If Caucasian cardiologists perform provocative testing with pacemakers similar to Japanese cardiologists, the prevalence and incidence of CAS may be higher than ever thought. However, in Taiwanese specialists’ experiences without implementing pacemakers when performing provocative testing using the bolus injections of ergonovine, there has been no cardiac arrest but only rarely mild bradycardia, which can quickly return to normal after immediate intracoronary administration of nitroglycerin once CAS occurs [63,64]. Fifth, the definition of positive epicardial CAS is different among previous Japanese, Taiwanese, and Caucasian studies [60]. For example, the definition of provoked CAS is a reduction of >50% [65], >70% [61,66,67], >75% [57,68,69,70,71], >90% [24,51,71,72], or 99–100% [73] in luminal diameter compared with postintracoronary nitroglycerin. Sixth, among all the clusters of CAS risk factors, the predominant factors that cause CAS in Asian patients may be different from those in white patients. Notably, it is estimated that nearly 1 billion people globally, most of whom are Asians, carry the Glu504Lys polymorphism in the aldehyde dehydrogenase 2 (ALDH2) gene [74]. This ALDH2 mutant is significantly associated with a high level of high-sensitivity C-reactive protein [75], which is a risk factor for CAS. In conclusion, while previous studies demonstrate that existence of racial heterogeneity in coronary vasomotor response [50], the prevalences of CAS and CAS-related HF (CASHF) in different populations are largely unknown.

In the US, while the incidence and prevalence of HF are increased [11,76], the age-specific incidence of HF might be reduced, but to a lesser degree in HFpEF compared with HFrEF [77]. A UK study showed that the age-adjusted incidence of HF fell by 7% between 2002 and 2014, whereas the absolute incidence of HF increased by 12%, and prevalent HF increased by 23% [78]. This growth in the absolute number indicates population aging, reduced mortality from cardiovascular diseases, including MI [78], and the increasing prevalence of risk factors. Given that approximately 50% of HFrEF cases can be attributed to ischemia [79], a new diagnosis of HFrEF frequently needs an assessment for underlying CAD. Despite the fact that individual-level factors (eg, old age, serious comorbidities, non-candidates, or no preference for coronary revascularization) should be evaluated before referral, coronary angiography remains the gold standard for diagnosis of obstructive CAD [80]. In addition to epicardial CAD, microvascular CAD is becoming widespread and often under-recognized [81]; hence, both epicardial and microvascular CAD, clinically overt or silent, acute or chronic, can result in decreased perfusion, myocardial damage, and further reduced myocardial function.

## 3. Clinical Features of CASHF

Until recently, patients with non-obstructive CAD were often inappropriately reassured due to assuming a favorable prognosis without further investigation, although clinical features may require coronary angiography. However, ischemia with non-obstructive CAD (INOCA) is a non-benign condition correlated with an equivalent incidence of adverse events as well as poor quality of life compared to obstructive CAD [82]. INOCA is defined as when patients present with symptoms and signs suggesting ischemia but are found to have no obstructive CAD at coronary angiography [83]. Indeed, this phenomenon is a primary cause of myocardial ischemia and is linked to a high risk of MI, decompensated HF, stroke, and unexpected sudden death [84,85,86,87,88,89,90]. On the other hand, CAS-related angina is common [27,91], although not as frequent as classic Heberden’s angina. While in the Taiwanese general population, the prevalence of CAS and obstructive CAD over 12 years of follow-up is 0.067% and 8.7%, respectively [92], the prevalence of CAS in other racial populations needs to be clarified and the frequency of CAS diagnosis might be increased when careful criteria are applied for its detection [91]. In Japan, although non-invasive provocation tests such as hyperventilation tests and cold pressor tests decreased remarkably in 2014, diagnosis by invasive provocation tests and the occurrence of CAS-related angina increased in 2014 compared with 2008, albeit not significantly [93]. CAS provocation tests using left-to-right coronary sequential evaluation were employed in just 30% of the Japanese hospitals; hence, although 40% of the centers were dissatisfied with standard spasm provocation tests, the majority of the hospitals confirmed the necessity of CAS provocation tests in the future [93]. As a result, considering that (1) CAS can cause resting angina with S-T segment depression and/or pseudonormalization of T waves; (2) asymptomatic CAS-induced angina is common [91,94]; and (3) cold-induced angina may arise from CAS [95], the incidence of CAS-induced myocardial ischemia could be doubtlessly much higher than that of the public perception indicated by the clinical presentation.

In clinical practice, angina pectoris results from a temporary myocardial oxygen-supply-demand imbalance [96,97], causing 2 types of ischemia, exertional demand ischemia and non-exertional supply ischemia [98]. Because coronary disorders are dynamic [99], non-obstructive CAD might become flow-limiting stenosis if the vascular tone is increased [99], and hence increased oxygen demand might not always precede myocardial ischemia [100]. In this regard, endothelial dysfunction makes up about two-thirds of symptomatic patients of INOCA and a smaller percentage of “MI with non-obstructive CAD” (MINOCA). The clinical spectrum of INOCA includes epicardial CAS, microvascular CAS, or mixed epicardial/microvascular CAS [19]. Microvascular CAS can cause myocardial necrosis, mild elevations of cardiac troponin, subtle left ventricular contractile abnormalities [101], and early-stage HF [102]. Previous studies showed that microvascular CAS could be demonstrated not only in angina patients with normal epicardial coronary arteries but also in HFpEF [4,103,104,105]; hence, angina and dyspnea can appear at 2 extremes in the presence of a continuum of disease contributing to the development of microvascular CAS and HFpEF [106]. Furthermore, 30–48% of in-patients receiving the optimal treatment for HFrEF <45% have provoked epicardial CAS [23,24,67,72]. While the prevalence of hypertension and smoking are higher in epicardial CAS-related than non-CAS-related HFrEF [24], more research is required to evaluate the risk factors of epicardial CASHFrEF.

Various arrhythmias, especially ventricular premature complex, more often appear in >50% of CAS-induced angina than in classic Heberden’s angina pectoris [39,107], albeit through unknown mechanisms but possibly involving QT dispersion in CAS-induced cardiac arrest and syncope [108]. While the severity of CAS has no relationship with the occurrence of these arrhythmias, ventricular arrhythmias occur more frequently during anterior wall ischemia [109]; however, right CAS-induced ventricular arrhythmias are not uncommon (Figure 2). Besides, ventricular fibrillation (VF) complicating CAS is responsible for sudden death with morphologically normal coronary arteries in autopsies, as previously reported [110]. Although cardioversion is always required to terminate VF, VF induced by CAS, epicardial or microvascular, rarely terminates spontaneously [50,80] (Figure 3 and Figure 4). Additionally, in a study of patients who had implanted cardioverter defibrillators, VF was asymptomatic in 43% and nonsustained in 40% of episodes [111]. The probability of syncope or pre-syncope is 25% and 62% when VF is <10 and ≥10 s, respectively [111]. On the other hand, about 40% of CAS-related inferolateral J wave and VF does not cause angina at the first VF, and could have been misinterpreted as early repolarization syndrome [112], which in the younger age group is associated with features of CAS such as lower systolic blood pressure and lower heart rate [113]. Therefore, CAS is essential and should be included in the differential diagnosis of syncope and early repolarization syndrome, prompting optimal medical management.

On the other hand, extreme CAS may cause life-threatening pulseless electrical activity or asystole without the occurrence of ventricular tachycardia or VF [50], which, when involving all of the three epicardial coronary arteries, can suddenly stop heartbeat due to pulseless electrical activity and flash freeze the whole myocardium immediately, leading to invisible coronary flow [114] and, despite intracoronary administration of nitroglycerin, prolonged contrast retention in the coronary arteries. While extended continuous cardiac massage is effective to resolve CAS-related pulseless electrical activity [114], cardiac pacing or implantable cardioverter defibrillator may not be feasible for the recovery of viable muscle from frozen myocardium during pulseless electrical activity, and may result in unexplained death [114,115]. Furthermore, ischemia of the sinus node or atrioventricular node arteries due to CAS can affect the development of pulseless electrical activity or asystole [114]. Taken together, without the induction of ventricular arrhythmias, CAS can directly result in pulseless electrical activity or asystole.

## 4. Pathogenesis

CAS not only causes remarkable progression of CAD but also demonstrates a “jump-up” phenomenon in as short as 25 min in a swine model [116], in which CAS superimposed on minimal coronary stenosis can rapidly progress to total atherosclerotic obstruction, resulting in MI [117]. In a pig model, the phenomenon can be explained by that CAS of abrupt rather than gradual onset can cause intramural hemorrhage in the plaque’s neovasculature and the subsequent sudden progression of organic coronary stenosis, leading to MI [118]. Most importantly, while smoking, age, C-reactive protein (CRP) [50], ALDH2 deficiency [119], and lipoprotein(a) [120] are risk factors for CAS, CAS is not associated with the classic risk factors for CAD [50,121], such as diabetes mellitus, hypertension [53], hypercholesterolemia [50,121] and obesity [50,121], suggesting pathophysiological differences exist between CAS and CAD. While risk factors in an individual usually exist together and have a cumulative and interactive effect to increase a person’s chance of getting CAS (Figure 5), precipitating factors refer to a specific event, which may act in the same patient to cause the onset of CAS in various circumstances. Notably, while older rather than younger people are more likely to develop CAS, smoking in the younger compared with their older analogs has a more powerful effect on CAS occurrence [122]. In addition, as smoking and age appear to have a more important role in men [123], CAS risk factors may be gender-specific.

The relaxation and contraction of vascular smooth muscle cells are regulated primarily through dephosphorylation and phosphorylation of the myosin light chain, respectively. CMD is characterized by impaired microvascular smooth muscle cell dilation, which can ultimately lead to HFpEF [124]. In addition, elevated Rho-kinase activity of smooth muscle cells favors contraction by directly increasing sensitization of the myosin light chain to Ca^2+^ and indirectly augmenting phosphorylation of the myosin light chain [125]. The Rho-kinase activity in vascular smooth muscle cells is elevated after wrapping the coronary arteries with interleukin (IL)-1β beads in a pig CAS model [125,126,127]. Other animal models of spontaneous CAS include KATP mutant or SUR2 KATP knockout mice, suggesting that loss of function of KATP channels can induce hypercontraction of smooth muscle cells without atherosclerosis [128,129]. Mice lacking α1H T-type calcium channels show a normal contraction of coronary arteries but decreased response in acetylcholine-induced relaxation [130]. Together, these models reveal that hyperreactivity of vascular smooth muscle cells can cause CAS via various pathways, whereas their clinical relevance in humans remains largely unknown.

The majority of CASHF patients present with a less severe phenotype (stages A and B) than overt HF, among which stage A is for patients at risk for HF but without current or prior symptoms or signs of HF and structural or biomarker evidence of heart disease, and stage B is for patients without current or prior symptoms or signs of HF but evidence of structural heart disease or abnormal cardiac function, or elevated natriuretic peptide levels [131], underscoring the importance of understanding risk factors for CAS to prevent the future burden of HF. Among the risk factors of CAS [132], smoking (relative risk 1.47) is independently associated with incident HF [133]. While no circulatory factor impeding oxygen supply to the heart such as fixed coronary stenosis or exercise is responsible for eliciting CAS-related angina, from the onset of the electrocardiographic abnormalities to the start of their reversion, the mean heart rate and arterial blood pressure are decreased, and isovolumic contraction time is lengthened, reducing the left ventricular performance [134]

Inflammation has been shown to account for the dissimilarities in cardiac remodeling between HFpEF and HFrEF. While HFpEF is linked to concentric hypertrophy, adverse remodeling in HFrEF is often due to ischemia-induced progressive loss of cardiomyocytes, with a patchy distribution of replacement fibrosis of dead cells by collagen, leading to LV dilatation and maladaptive remodeling [135,136,137,138]. Furthermore, because pathophysiological differences exist between HFpEF and HFrEF, the inflammatory biomarkers, including CRP and IL-6, are higher in HFpEF than in HFrEF, while markers of cardiomyocyte injury, such as high-sensitivity troponin T and brain natriuretic peptides, are higher in HFrEF than in HFpEF [139].

Chronic myocardial dysfunction resulting from hypoperfusion, hibernation, or both, may also increase the risk of HF [140,141]. Subjects with both epicardial CAD and CMD may have chronic hypoperfusion-associated inflammation and fibrosis, resulting in increased myocardial stiffness. Similarly, episodic coronary hypoperfusion such as CAS may cause myocardial functional impairment for hours to days (myocardial stunning) [142]. Studies with positron emission tomography [143] and single photon emission computed tomography [144] show decreased blood flow and glucose uptake in myocardial areas that concomitantly have decreased systolic function. In addition, ventricular diastolic dysfunction appears in both experimental [145] and clinical ischemia [146]. Of note, the decrease in coronary vasodilator reserve is proportional to extent of arterial luminal stenosis [147,148]. Consequently, myocardium with normal resting blood flow may have decreased exercise blood flow and may display decreased glucose metabolism on positron emission tomography during exercise and a concomitant decrease in ventricular function.

As associated researchers at the National Human Genome Research Institute unlock the mysteries of the complete set of the human genome, almost every disease has a genetic component [149], which is the case with CAS in that a mutation in the ALDH2 gene is believed to be the cause [150] and associated with Asian flush syndrome. Mizuno Y et al., showed that Asians with defective ALDH2*2 alleles have a higher risk of CAS. They also found that the defective gene positively interacts with the detrimental effects of smoking on stronger vasoconstriction than each factor alone by increasing reactive aldehydes [151]. Furthermore, other Japanese studies show that the mutant ALDH2*2 allele carriers compared with subjects with the ALDH2*1/1 genotype have higher frequencies of more severe CAS-related myocardial injury [119,152].

The mechanisms of coronary vasomotor disorders can be endothelium-dependent or endothelium-independent [19]. While endothelium-dependent dysfunction results from an endothelium-derived disparity between relaxing factors, e.g., nitric oxide (NO), and constrictors, e.g., endothelin [19], endothelium-independent function relies on vascular myocyte tone [19]. Endothelial dysfunction, as in CAS, reduces the bioavailability of nitric oxide (NO), cyclic guanosine monophosphate, and protein kinase G in adjacent cardiomyocytes [153], contributing to myocardial fibrosis and HFpEF [154,155]. While during HF development, the initial inflammatory response is a protective reaction to tissue injury, it may lead to irreversible damage when the inflammation is prolonged. Pathologic features, common to all cardiomyopathies irrespective of origin, include ventricular hypertrophy, fibrosis, scarring, and dilatation [102]. This phenomenon was investigated in 2 animal models of congestive cardiomyopathy: the hereditary cardiomyopathic Syrian hamster and the hypertensive-diabetic rat [102]. In both the genetic and the acquired disease models, there was focal myocytolytic necrosis with the subsequent healing with focal scars, ventricular hypertrophy, ventricular dilatation with congestive HF, and, finally, death [102]. In both diseases, the microcirculation of the animal hearts had been studied by the use of silicone rubber perfusions; microvascular CAS was demonstrated early in the disease associated with small areas of myocytolytic necrosis and subsequent fibrosis [102]. Because the distance between cardiomyocytes and endothelial cells is fewer than 3 μm [19], allowing for sufficient blood supply and bidirectional influences, both myocardial fibrosis- and hypertrophy-induced subendocardial ischemia may cause left ventricular diastolic dysfunction and longitudinal systolic abnormalities, leading to remodeling and HFpEF, which may reciprocally trigger subendocardial ischemia and endothelial dysfunction in return [156]. Collectively, although myocardial ischemia directly contributes to HFpEF [157], the causes and mechanisms contributing to HF as well as CASHF, albeit largely unknown, are likely multifactorial (Table 1).

### 4.1. Microvascular CASHF

Several cardiac and systemic disorders, such as HFpEF, brain small-vessel disease, diabetes, hypertension, chronic inflammatory and autoimmune diseases, and chronic kidney disease can develop INOCA [198,199]. In most of these patients, close relationships exist between microvascular dysfunction and atherosclerotic epicardial CAD [198,199]. In early diabetic rats, the coronary microvascular focal and segmental constrictions occur when prostacyclin and nitric oxide production is prevented, which, if left untreated in advanced diabetes, will progress to irreversible microvascular damage [200]. On the other hand, while CMD does not develop atheroma in accord with epicardial atherosclerosis, coronary microcirculation in patients carrying cardiovascular risk factors can evolve into structural and functional atherosclerotic-like changes [19], presenting as either vasodilator abnormality and/or microvascular CAS [19]. Furthermore, CMD can occur in the absence or presence of obstructive epicardial CAD [201]. As a result, the CMD-related myocardial ischemias are unlike those attributable to epicardial flow-limiting stenosis, in which the regional ischemia perfused by the obstructed epicardial artery is homogeneously distributed, resulting in regional wall motion abnormality [19]. In contrast, myocardial ischemia in CMD may appear as patchy and not entail all microvessels originating from an epicardial artery, causing symptoms without wall motion abnormalities [202], or as a generalized phenomenon resulting in diffuse perfusion and wall motion abnormalities, and thereby HFrEF with a normal result on noninvasive stress imaging tests.

Circulating factors, such as fibrocytes, circulating monocyte-derived cells, and fibroblasts, might regulate the effects of microvascular CAS favoring the development of left ventricular fibrosis and hypertrophy [203]. While fibrocytes are recruited to chronically injured myocardium in cardiac remodeling in mice, treatment with serum amyloid P decreased fibrocyte accumulation and fibrosis [204]. Another modulatory factor, atrial natriuretic peptide, may induce phosphorylation of Smad proteins, thus inhibiting their nuclear translocation and binding to TGF-Smad responsive elements in the promoter regions of extra-cellular matrix genes [205]. An auxiliary potential mechanism, as proposed by Pepine et al. [206], involves repetitive cycles of ischemia-reperfusion such as in CAS that impede cardiac myocyte relaxation thereby causing diastolic dysfunction and HFpEF.

In addition, endothelial dysfunction associated with inflammation reduces the content of cyclic guanosine monophosphate (cGMP), protein kinase G (PKG), and transforming growth factor (TGF)-β in cardiomyocytes and microvascular NO bioavailability, all of which are involved in the physiological modulation of cardiac hypertrophy and stiffness [207,208,209,210] Besides, NO reduction inhibits cGMP and TGF-β functions, favoring conversion of endothelial cells into mesenchymal cells such as fibroblasts [210,211,212]. Overall, these changes promote hypertrophy, fibrosis, and the subsequent development of left ventricular diastolic dysfunction.

### 4.2. Epicardial CASHF

Although epicardial atherosclerosis may induce endothelial dysfunction of CMD, atherosclerotic CMD may in reverse accelerate the development of epicardial atherosclerosis through decreased blood flow and wall shear stress, leading to progressive epicardial endothelial dysfunction [213] and thrombus formation [214]. Similarly, CAS involving epicardial and microvascular arteries is therefore considered the expression of the same CAS development sharing a common pathophysiological milieu that affects the entire coronary circulation [215]. A substantial body of evidence suggests that subjects with microvascular angina have 2 important extra features contributing to angina symptoms: (1) hyperreactivity of smooth muscle cells to microvascular constrictor stimuli; (2) enhanced awareness of cardiac pain-provoking stimuli. Indeed, a significant number of patients with microvascular angina have microvascular CAS, which is angina accompanied by ST-segment depression after the intracoronary acetylcholine provocative testing [106].

A previous study using substance P, a pure endothelial-dependent vasodilator, demonstrates that in patients with variant angina, endothelial dysfunction at sites of CAS is not necessarily present [216]. Furthermore, in variant angina, several studies fail to show endothelial dysfunction in non-CAS coronary arteries as well as in peripheral arteries [217], and other studies also did not show the higher prevalence of NO synthase polymorphisms-associated endothelial dysfunction [218]. Altogether, an impairment of endothelium-mediated vasodilation appears unlikely to cause CAS by itself, although it might facilitate the effects of coronary vasoconstrictors “CAS prone” individuals [219].

CAS, particularly multi-focal spasms [23], causes myocardial necrosis via reperfusion injury [220], leading to reduced diastolic relaxation during angina [221], and the subsequent development of HFpEF and HFrEF [23,24,25,67,72]. Of note, left ventricular dysfunction may recover in about 2 min to baseline when the electrocardiographic abnormalities start returning to the pre-CAS state [36,134].

Takotsubo cardiomyopathy is an acute and reversible form of unexpected physical and emotional distress-related HFrEF featuring symptoms and signs of acute MI without CAD, in which the apex of the left ventricle balloon enlarges to resemble a takotsubo, a Japanese octopus pot [222]. Despite the syndrome more frequently occurring in older women than in men [222], it can affect people of any age, including a newborn after delivery distress with catecholamine-mediated cardiac toxicity [223]. Precipitating mechanisms are multifactorial and complex, including microvascular and epicardial CAS [222], genetics, and thyroid disorders [224]. Stress activates the sympathetic nervous system to release circulatory catecholamines and the hypothalamic-pituitary-adrenal axis to release circulatory glucocorticoids [225]. While initially protective for the heart, glucocorticoids not only increase plasma levels of catecholamines by inhibiting uptake but also induce cardiac supersensitivity to catecholamines, leading to an enhanced β-adrenoceptor signal transduction system [225]. Excessive catecholamines induce diminished apical and enhanced basal wall motion of the left ventricle due to the apicobasal adrenoceptor gradient [224]. Furthermore, low catecholamine levels stimulate cardiac Ca^2+^ movements, whereas excessive catecholamine levels induce intracellular Ca^2+^ overload in cardiomyocytes, resulting in cardiac dysfunction [226]. On the other hand, under stressful conditions, high catecholamine levels are oxidized to form oxyradicals, which can cause CAS [225]. Few cases of Takotsubo cardiomyopathy due to an angiographically confirmed focal, single vessel, or multivessel CAS have been reported. A retrospective analysis in 10 of 48 (21%) Takotsubo cardiomyopathy cases have shown positive provocative CAS, 5 of whom involved both right and left coronary arteries [227]. Angelini reported 4 cases of Takotsubo cardiomyopathy in which echocardiographic apical ballooning or similar symptoms could be reproduced by provocative CAS [228]. Moreover, it has also been demonstrated that alternate recurrent CAS and Takotsubo cardiomyopathy can exist in the same individual [229]. These observations underscore the importance of CAS as a culprit process underlying Takotsubo cardiomyopathy and the targeted treatments accordingly. Further studies will provide critical insights into this unique issue.

### 4.3. Cellular and Animal Models of Takotsubo Cardiomyopathy, CAS, and Microvascular CASHrEF

Since the late 1800s, because of the similarity in disease processes among animals and humans, animal models began to be developed and help elucidate the connection between dietary cholesterol and atherosclerotic progression [230]. Since then, the inflammatory and immunological nature of atherosclerosis has been revealed by several studies in patients and experimental models, underscoring the importance of inflammation in CAD, as well as in CAS. Investigation of disease-modifying mechanisms in these models will be crucial for developing future diagnostics and therapy against CAS as well as CASHF. We provide an introduction to experimental models that are used for CAS studies and the research techniques that can be utilized (Table 2). Whether these models can be used for CASHF experiments remains to be elucidated.

It stands to reason that the investigator must acknowledge the limitations of animal models so as to construct and interpret relevant experiments cautiously when extrapolated to humans. Problems in cross-species extrapolation and local differences in the arteries are well known. The response of the coronary artery in dogs appeared to be different from that in humans [116]. Isolated vessels often do not respond in the same way in vitro as in situ, even in the same species. Although, often, only part of the disease is triggered in the animal during an experiment, even studying these partial processes may help understand the course and mechanisms of a disease. Genetically modified mice are playing an increasingly important role in this type of research.

## 5. Treatment

Traditionally, unless contraindicated, HFrEF should be treated with β-blocker, angiotensin receptor–neprilysin inhibitor, angiotensin-converting enzyme inhibitor, or angiotensin receptor blocker, with the addition of a mineralocorticoid receptor antagonist in patients with prominent symptoms [80], while Ivabradine and hydralazine/isosorbide dinitrate may also be considered in the management of HFrEF [249]. More recently, sodium-glucose cotransporter 2 inhibitors have much-improved disease outcomes, dramatically reducing cardiovascular and all-cause mortality regardless of diabetic state, and vericiguat, a stimulator of soluble guanylate cyclase, reduces inpatient admissions for high-risk HFrEF [80]. Device therapies may have benefits for specific subpopulations of HFrEF [80]. On the contrary, medication classes that are efficacious in HFrEF have been less so in HFpEF, decreasing the risk of inpatient admissions but not cardiovascular or all-cause mortality in HFpEF [249]. These observations underline the significance of non-cardiac comorbidities and underscore the complexity of pathophysiological mechanisms, both cardiac and non-cardiac, underpinning HFpEF [249].

CASHF cannot be improved by interventional revascularization, and medications are the cornerstones of treatment. Although the interplay of epicardial and microvascular CAS and associated risk factors are clinically relevant and represents a critical differentiator for what may constitute specific therapeutic strategies in CASHF, the CAS-induced abnormal regional wall motion, dilated left ventricular and reduced systolic function improved 6 months to >1 year by medications, including calcium channel blockers and nitrate/nicorandil [23,72]. HFrEF, e.g., dilated cardiomyopathy in Syrian hamsters [220,250] and in German patients [251], with CMD possibly caused by CAS, can be improved through vasodilator effect by the medical treatment with verapamil and diltiazem, respectively. Of note, while auxiliary diltiazem in suspected CAS-related dilated cardiomyopathy has mortality benefits, improved symptoms, and hemodynamics by reducing afterload, arrhythmias, and catecholamine levels [251], diltiazem in individuals with infarction-related HFrEF has a dismal prognosis [252]. On the other hand, calcium channel blockers in non-ischemic HFrEF are not recommended as first-line therapy. Hence, although first-generation calcium channel blockers (except amlodipine and felodipine), dihydropyridine, and nondihydropyridine, should be limited in non-CAS-induced HFrEF because of no functional, mortality, or outcome benefits [253], if HFrEF patients have provoked CAS, calcium channel blockers might improve myocardial ischemia due to CAS [23,67]. Future research is needed to investigate the potential therapeutic role of calcium channel blockers in CASHFrEF. In contrast, the use of β-blockers in CASHFrEF may aggravate CAS [25]. Finally, although fasudil, a Rho-kinase inhibitor, prevents acetylcholine-induced CAS and associated myocardial ischemia [254], its role in CASHF remains unknown. Additionally, patients with CAS-related dilated cardiomyopathy have a higher prevalence of atrial fibrillation than those without CAS [67% vs. 8% (*p* < 0.05)] [67]. Therefore, dilated cardiomyopathy with atrial fibrillation is probably an indication to identify CAS [67]. Taken together, although no guideline addresses the therapeutic significance of calcium channel blockers in CASHF [24], the differential diagnosis of dilated cardiomyopathy or HFrEF should include CAS since calcium channel blockers are potentially promising medical options [24,72].

Notwithstanding established treatments for CAD, some patients suffer from refractory symptoms. The soluble guanylate cyclase stimulator riociguat, licensed for pulmonary hypertension treatment, has been reported to resolve recurrent and refractory CAS-induced angina [255]. The drug inhibited the acetylcholine provocation of epicardial CAS, and resulted in a remarkably satisfactory long-term (10 months) effect on perceived well-being [255], suggesting that the soluble guanylate cyclase pathway is a potential novel therapeutic target in CAS. However, randomized controlled clinical trials are necessary to strengthen this presupposition.

In acute MI, although prompt coronary reperfusion is the most effective way to limit myocardial injury, the subsequent cardiomyocyte apoptosis, adverse left ventricular remodeling, and, finally, ischemic HF, the medical therapies of the subsequent tissue inflammation and its following suppression and resolution, remains largely unknown [171]. Among dietary phytochemicals that are naturally plant-derived and have been investigated to offer some protection against chronic diseases, garcinol demonstrates potential drug treatment effects in in vitro studies, such as its anti-inflammatory, anti-oxidative, and anti-cancer properties [256]. In rat models with isoproterenol-induced HFrEF, garcinol treatment increased the heart rate and improved the maximum rate of increase in pressure (+dp/dtmax), maximum rate of decrease in pressure (−dp/dtmax), ejection fraction, and systolic pressure in the left ventricle [257]. We have previously demonstrated that garcinol suppresses lipoprotein(a)-induced oxidative stress and inflammatory cytokines by α7-nicotinic acetylcholine receptor-mediated inhibition of p38 MAPK/NF-κB signaling in cardiomyocyte AC16 cells and isoproterenol-induced acute MI mice [258]. These observations suggest that garcinol may effectively prevent cardiomyocyte apoptosis.

Although inflammation can be a cause–effect event of HF and, hence, a therapeutic target, clinical trials evaluating anti-inflammatory treatments failed to produce adequate relief; however, it is as yet uncertain what targeted anti-inflammatory therapy in distinct sub-phenotypes of HF such as CASHF will prove to be successful [259]. Potential targeted anti-inflammatory therapies include the inhibition of IL-1β, IL-6, and galectin-3 [259]. In CANTOS (Canakinumab Anti-Inflammatory Thrombosis Outcomes Study) trial, subjects who were canakinumab-responsive (as reflected by a decrease in C-reactive protein) had a significant drop in HF hospitalizations and the composite of HF hospitalizations and all-cause mortality by 38% and 32%, respectively, compared with placebo [260]. Although anti-IL-6 therapies have been approved for rheumatologic and inflammatory disorders, including tocilizumab, siltuximab, and sarilumab, no clinical trial yet investigates the effects of IL-6 inhibitors in HF patients [259]. In experimental preclinical studies, pharmacologic inhibition of galectin-3 through the utilization of either modified citrus pectin or N-acetyllactosamine avoids myocardial and renal fibrosis and dysfunction [261,262], which may warrant further investigation in HF. Among anti-inflammatory agents, NSAID use has previously been linked to an increased risk of hypervolemia, blood pressure elevation [263], and HF [264]. In large-scale clinical trials, anti-TNF-α agents did not prevent HF [265]. Taken together, to date, clinical trials of directed anticytokine and anti-inflammatory treatments for HF have proved mostly unsuccessful [259], and the effects of these therapies have yet studied in CASHF. Of note, because cytokines can become cardioprotective in certain conditions, the timing (acute or chronic phase following MI) and intensity of the cell type-specific inhibition (leukocytes, cardiac fibroblasts or cardiomyocytes) must be taken into account in developing anti-inflammatory therapies [169].

## 6. Conclusions

CAS is common, though it is still unsolved, and deserves the same fast action as CAD. Because CAS can cause rapid plaque progression of CAD and the development of acute coronary syndrome, including MI, underrecognized CAS is concerned with the health of the individuals and population as a whole, as well as with the health implications of the economic and social policies, and investment in health policies. While medical care can prolong survival and improve prognosis after the occurrence of CAD and HF, more important is to identify ill people afflicted with CAS before the potential subsequent development of CAD and HF in the first place. It has been demonstrated that after the HF condition is stabilized, the provocative testing for CAS can be safely performed. Treatment should be started early once CAS is diagnosed. On the other hand, while HF treatment aims to control the symptoms and slow down the progression, CASHF is one of a few conditions in that medical therapy may reverse HF. Furthermore, patients with CASHFrEF may have associated atrial fibrillation.

CAS has been a multifactorial disorder that cannot be attributed to a single factor alone (Figure 6). In addition, because vascular smooth muscle cell hyperreactivity is a nonspecific reaction and CAS-induced angina is not improved by rest, CAS-related angina can occur under different situations in the same patient. As a consequence, identifying CAS is crucial in clinical practice because the therapeutic strategies between CAS and obstructive CAD are different, and calcium channel blockers are needed to improve the left ventricular function of CASHFrEF. Accordingly, it is of paramount importance to administer intracoronary nitroglycerin adequately before coronary interventions to distinguish spontaneous CAS from obstructive CAD, thus limiting vascular damage to all layers and preventing unnecessary interventions. Finally, we agree with the Japanese cardiologists to recommend upgrading the pharmacological CAS provocative testing to Class I in the guidelines in patients with angina but without obstructive CAD throughout the world.

Avoiding cigarette smoking and alcohol with concomitant appropriate dosing and timing of calcium antagonists remain the mainstay of CAS therapy. Besides, instead of treating a specific physical condition, we should focus on the whole person’s health. Anxiety and depression confer high risks for CAS-related myocardial ischemia. In Taiwanese patients, anxiety is associated with a remarkably 5-fold increased risk of incident CAS [92], suggesting that simple assessment tools can be used for patients at risk for CAS to evaluate mental health well-being, and treatments of psychological disorders can have a beneficial impact on CAS [266]. Because recurrent angina events are commonly observed in CAS, further investigation is needed and important to help better clarify the responsible molecular mechanisms and manage CAS more effectively.

## Figures and Tables

**Figure 1 ijms-24-07530-f001:**
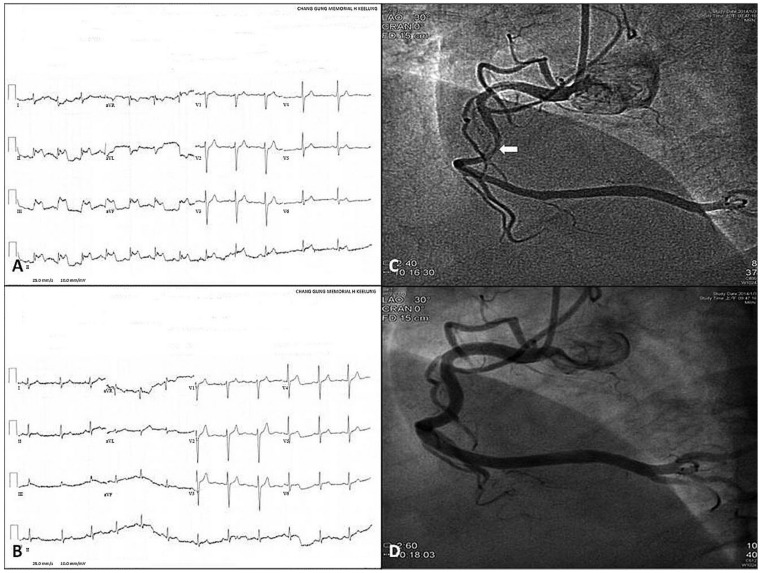
The 12-lead electrocardiograms and coronary angiography of variant angina. Angina attack (**A**) and post-sublingual nitroglycerin 0.6 mg (**B**) 12-lead electrocardiograms of a 47-year-old male revealed brief ST-segment elevation in II, III, and aVF leads. Ten months later because of recurrent chest pain, he underwent coronary angiography. The coronary angiography revealed intracoronary methylergonovine-induced CAS in the middle portion of the right coronary artery (**C**, arrow), which was alleviated after intracoronary nitroglycerin 200 μg (**D**). (Reproduced from [33], with permission of the publisher.).

**Figure 2 ijms-24-07530-f002:**
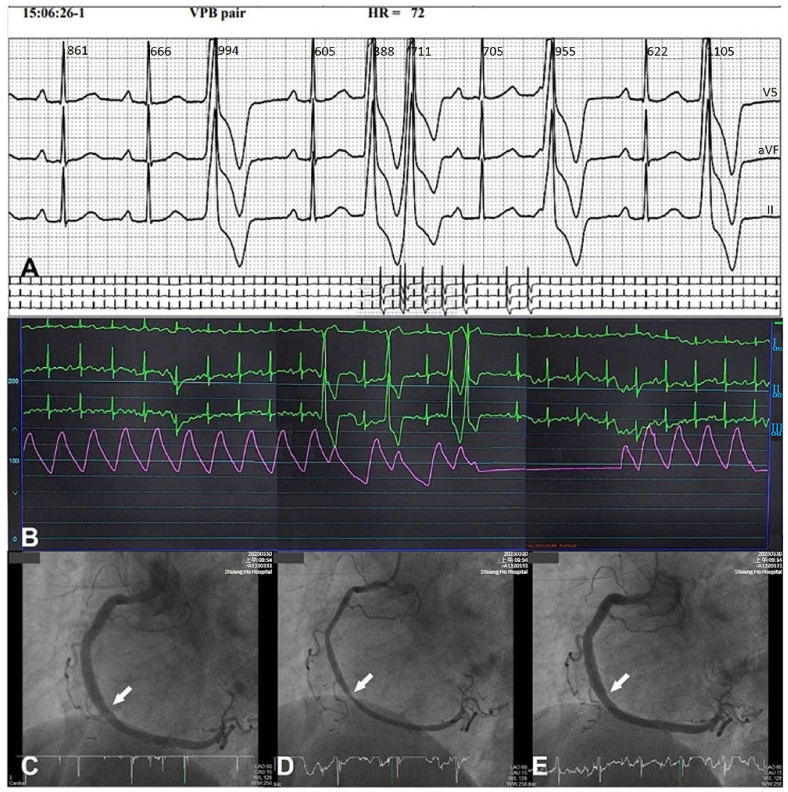
Epicardial CAS-induced ventricular ectopics: 24 h Holter monitor, electrocardiograms, pressure tracing, and right coronary arteriogram in a 53-year-old female presenting with frequent palpitation and unstable rest angina. (**A**) A 24 h Holter monitor showed sinus rhythm with runs of ventricular ectopics in singles and couplets without preceding ST segment changes; (**B**) simultaneous lead I, II, III electrocardiogram and systemic arterial pressure tracing during intracoronary ergonovine testing; (**C**) baseline angiographically normal right coronary artery with minimal plaquing; (**D**) middle spasm (arrow) immediately after intracoronary administration of 45 μg ergonovine. Ventricular ectopics in singles and one couplet occurred at the same time; (**E**) the CAS and ventricular ectopics were relieved after intracoronary administration of 200 μg nitroglycerin. The patient’s consciousness remained clear throughout the examination.

**Figure 3 ijms-24-07530-f003:**
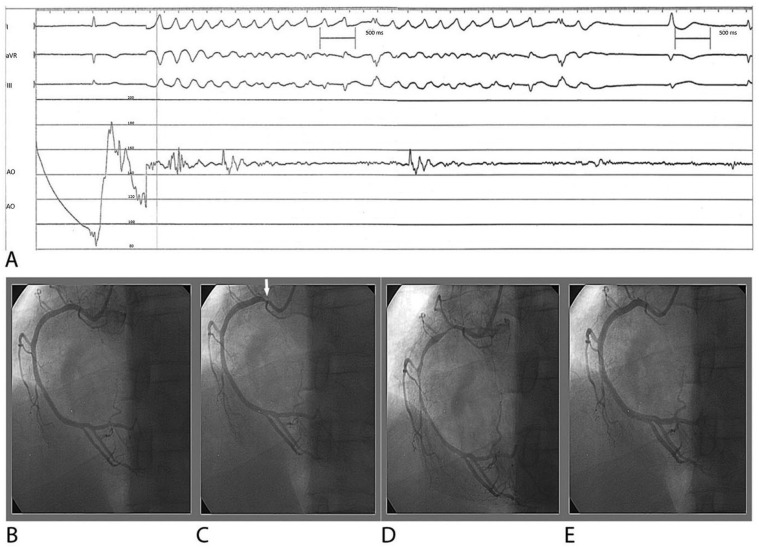
Epicardial CAS-induced VF: electrocardiograms, pressure tracing, and right coronary arteriogram in a 50-year-old male with unstable angina, presenting after wakening with rest angina at night. (**A**) Simultaneous lead I, II, aVR electrocardiogram, and systemic arterial pressure tracing during intracoronary ergonovine testing; (**B**) baseline angiographically normal right coronary artery with minimal plaquing; (**C**) ostial spasm (arrow) immediately after intracoronary administration of 15 μg ergonovine; (**D**) in 10 s, the ostial spasm recovered spontaneously, multi-focal spasms appeared in the proximal and middle portion, and ventricular fibrillation occurred at the same time for 10 s and recovered spontaneously without intervention; (**E**) multi-focal spasms were relieved after intracoronary administration of 100 μg nitroglycerin. The patient’s consciousness remained clear throughout the examination. (Reproduced from [63], with permission of the publisher).

**Figure 4 ijms-24-07530-f004:**
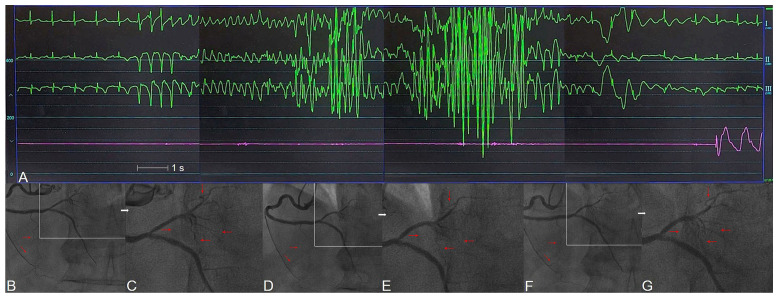
Microvascular CAS-induced VF: electrocardiograms (green line), pressure tracing (purple line), and right coronary arteriogram in a 75-year-old male with unstable rest angina. (**A**) Simultaneous lead I, II, III electrocardiogram and systemic arterial pressure tracing during intracoronary ergonovine testing; (**B**,**C**) baseline angiographically normal right coronary artery with minimal plaquing (white and red arrows); (**D**,**E**) microvascular spasm (white and red arrows) immediately after intracoronary administration of 45 μg ergonovine. Ventricular fibrillation occurred at the same time for 17 s and recovered spontaneously without intervention; (**F**,**G**) microvascular spasms were relieved after intracoronary administration of 200 μg nitroglycerin (white and red arrows). The patient’s consciousness remained clear throughout the examination.

**Figure 5 ijms-24-07530-f005:**
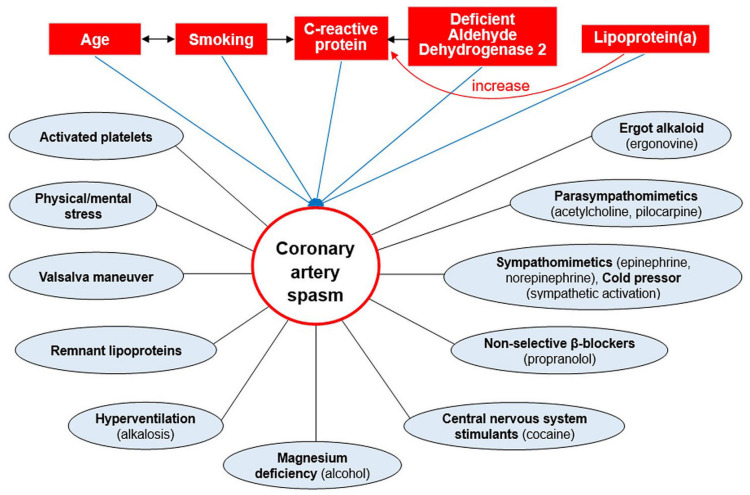
Risk factors and precipitating factors are represented by rectangles and circles, respectively, for CAS. (Adapted from [33], with permission of the publisher).

**Figure 6 ijms-24-07530-f006:**
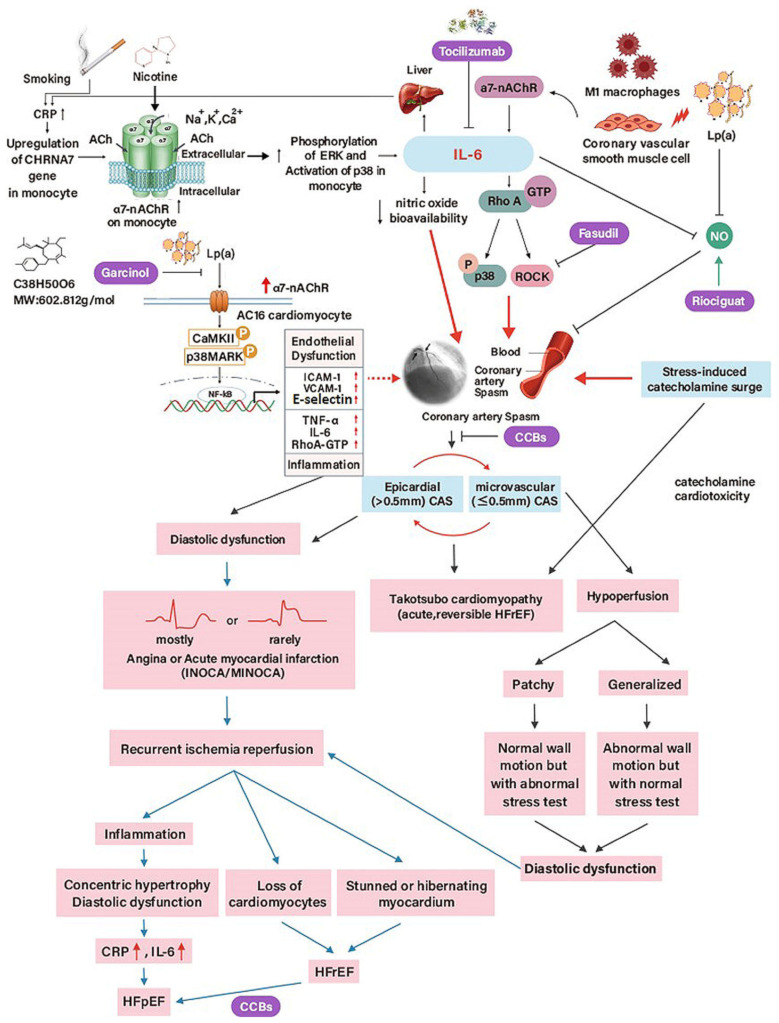
Graphical abstract depicting the multifactorial molecular and cellular mechanisms involved in the initiation and progression of CAS and CAS-related preclinical HF to clinical overt HF. The development of CAS can be contributed to by smoking, CRP, and Lp(a). Fasudil, Tocilizumab, Garcinol, and Riociguat are potential disease-modifying therapies of CAS [196,197,201,252]. The reversible nature of CASHF is suggested and represented by a reciprocal relationship and a positive feedback loop between epicardial and microvascular CAS. Solid arrows: direct activating interactions; Dashed arrow: indirect activating interactions; Blunt arrows: inhibition. Ach: acetylcholine; α7-nAChR: α7-nicotinic acetylcholine receptor; CHRNA7: α7-nAChR protein coding gene; CamKII: calmodulin-dependent kinase II; CAS: coronary artery spasm; CRP: C-reactive protein; HFpEF: heart failure with reduced ejection fraction; HFrEF: heart failure with reduced ejection fraction; ICAM-1: intercellular adhesion molecule 1; IL-6: Interleukin-6; INOCA: ischemia with non-obstructive coronary artery disease; Lp(a): lipoprotein(a); MINOCA: myocardial infarction with non-obstructive coronary artery disease; p38MAPK: p38 mitogen-activated protein kinase; VCAM-1: vascular cell adhesion molecule 1.

**Table 1 ijms-24-07530-t001:** Proposed mechanisms of coronary artery spasm-related heart failure.

Etiology	Mechanism
A. **Cardiomyocyte**	
Hypertrophy	(1) Cardiomyocyte apoptosis, necrosis, degeneration, and interstitial fibrosis are partially compensated by hypertrophy involving DNA synthesis and transcription [158]. (2) Activation of the Akt pathway with the subsequent inhibitory phosphorylation of glycogen synthase kinases-3β [159], and phosphorylation of calmodulin-dependent kinase II (CamKII) [159]. (3) Release of angiotensin-II and endothelin-1 [159].
2. Impaired excitation-contraction coupling	(1) Impaired calcium uptake by sarco/endoplasmic reticulum Ca^2+^-ATPase (SERCA) 2a (2) Uncontrolled calcium efflux through ryanodine receptors [160].
3. Inflammation	(1) During myocardial inflammation, while infiltrating macrophages in the myocardium are the major source of inflammatory cytokines, cardiomyocytes, and fibroblasts can also produce cytokines [161]. (2) Cardiomyocytes can transform into a proinflammatory and profibrotic secretory phenotype, forming a complex autocrine system between molecular pathways that lead to cell death [162]. (3) Increased paracrine secretion of tumor necrosis factor-α (TNF- α) via transcriptional regulation of activator protein-1 (AP-1) and nuclear factor-κB [163]. (4) Increased transforming growth factor-β (TGF-β) production [164], activating the renin–angiotensin–aldosterone system [159]. (5) Increased IL-6 production [165], but not IL-1β [166].
4. Fibrosis	(1) Increased synthesis of matrix metalloproteinases (MMPs) due to suppression of MMP inhibitors [167].
B. **Non-cardiomyocyte**	
Fibroblast	(1) During myocardial inflammation, while infiltrating macrophages in the myocardium are the major source of inflammatory cytokines, cardiomyocytes, and fibroblasts can also produce cytokines such as TNF-α and IL-6 [161]. (2) Under pathological conditions such as MI, fibroblasts are activated and differentiate into myofibroblasts in a large proportion, expressing features of smooth muscle cells such as contractile protein α-smooth muscle actin (α-SMA), [168]. Myofibroblasts are key sources of proinflammatory cytokines and the extracellular matrix and highly responsive to cytokines, including TNF-α, IL-6, and IL-1β [169]. (3) Deposition of type I and III collagen, extracellular matrix cross-linking, myocardial stiffness, and diastolic dysfunction [170], resulting in Impaired contractility, arrhythmias, local microfibrillations, and systolic dysfunction [171]. (4) Increased synthesis of matrix metalloproteinases (MMPs) due to the downregulation of MMP inhibitors [167].
2. Monocytes/Macrophages	(1) Monocytes/macrophages, fibroblasts, and cardiomyocytes all increaseTNF-α expression through different transcriptional regulatory systems including the activator protein-1 (AP-1)and NF-κB [163].
3. Endothelial cell	(1) Proinflammatory secretion [172] of IL-6 and TNF-α [173]. Most IL-1β production is localized to endothelial cells and interstitial macrophages rather than cardiomyocytes [166]. (2) Directly transdifferentiating into myofibroblasts in a small proportion [174], or performing endothelial-to-mesenchymal transition, generating cells that express endothelial markers while gaining fibroblast-like characteristics [174]. (3) Complement C1q tumor necrosis factor-related protein (CTRP)-9 overexpression to upregulate hypertrophy [175].
4. Lymphocyte	(1) Activation of fibroblasts in the myocardium [176].
5. Mast cell	(1) Activation of fibroblasts in the myocardium [177].
C. **Inflammation**	(1) A systemic inflammatory biomarker C-reactive protein has both diagnostic and prognostic values to predict the risk of developing HFpEF and subsequent cardiovascular events. However, previous results are conflicting [178]. (2) During myocardial inflammation, while infiltrating macrophages in the myocardium are the major source of inflammatory cytokines, cardiomyocytes, and fibroblasts can also produce cytokines [161]. (3) In HF, irrespective of left ventricular ejection fraction, inflammatory cytokines are consistently elevated [179], among which tumor necrosis factor-α, transforming growth factor-β and family of interleukins including IL-1β, -6, -12, -8, and -18 are the most greatly induced following cardiac damage [180]. (4) Inflammatory cytokines directly decrease contractility by inhibiting sarco(endo)plasmic reticulum Ca^2+^-ATPase 2a (SERCA2a) [181].
D. **Metabolism**	(1) Reduction of adenosine triphosphate and phosphocreatine concentrations, impairing fatty acid oxidation and mitochondrial carbohydrate metabolism [182]. (2) In the normal heart, the peroxisome proliferator-activated receptors (PPARs), mainly PPAR-α and PPAR-β, and their coactivator PGC-1α provide a more efficient pathway for aerobic energy production [183]. In ischemia [184], hypertrophy [185], and HF [186], the PGC-1α expression is decreased, resulting in decreased fatty acid utilization and increased glucose oxidation [187]. (3) PPARs-related PI3K/Akt pathway and its downstream targets, including glycogen synthase kinase-3β (GSK-3β), AMP-activated protein kinase (AMPK), and mammalian target of rapamycin (mTOR), are important in myocardial metabolism. Akt phosphorylation inhibits GSK-3β and AMPK activity, reducing energy production [188]. mTOR overexpression had decreased interstitial fibrosis in hypertrophy [189]
E. **MicroRNAs**	(1) Among the most examined miRNAs directly involved in cardiac fibrosis, miR-133a, miR-29, and the miR-21 families play a critical role [159,190].
F. **Mitochondria**	(1) Increased mitochondrial reactive oxygen species by angiotensin II through activation of ERK1/2 in mice, partly responsible for cardiac fibrosis and hypertrophy [191]. (2) Downregulation of genes involved in mitochondrial biogenesis, such as PGC-1α and PGC-1β, p38-mitogen-activated protein kinases (MAPK), and mitochondrial transcription factor A (TFAM), contributing to cardiac dilatation [192,193,194,195].
G. **Autophagy**	(1) Autophagy exerts cardioprotection in several cardiovascular diseases such as MI; however, prolonged activation may be detrimental [196], suggesting that the results of adaptive autophagic responses are determined by not only the autophagic intensity and duration but also other related signaling pathways.
H. **Apoptosis**	(1) Activated by ROS and upregulated downstream of angiotensin-II G-protein coupled receptor (GPCR) signaling, CAMKII mediates calcium dysregulation, triggering apoptosis [197].
I. **Genetics**	(1) The mutant ALDH2*2 allele carriers compared to subjects with the ALDH2*1/1 genotype have higher frequencies of more severe CAS-related myocardial injury [119,152].

**Table 2 ijms-24-07530-t002:** Proposed models of takotsubo cardiomyopathy and coronary artery spasm.

Models	Mimicry of Human Disease	Year	Author	Comments
A. **Cellular**				
Human Cardiomyocytes Derived from Induced Pluripotent Stem Cells (hiPSC-CMs)	Takotsubo cardiomyopathy	2022	Fan et al. [224]	Catecholamine-treated hiPSC-CMs or Takotsubo cardiomyopathy-specific iPSC-CMs mimic characteristics in line with those found in subjects with Takotsubo cardiomyopathy. Additionally, Takotsubo cardiomyopathy -iPSC-CMs provide a feasible and valid cell source for research of pathophysiological mechanisms, drug tests, ion channels, and gene functions.
			**Characteristics:** For the first time, after treatment with high levels of catecholamines hiPSC-CMs generated from 2 Takotsubo cardiomyopathy patients showed increased β-adrenergic signaling in iPSC-CMs [231].	
2. Patient monocyte-derived macrophage	CAS	2018 and 2022	Hung et al. [232] and Lin et al. [233]	
			**Characteristics:** (1) *Monocyte Isolation from Patient Peripheral Blood Mononuclear Cells:* After overnight fasting just before angiography, blood was collected in tubes and centrifuged for 20 minutes at room temperature. After removing the top layer without disturbing the red bottom layer, the opaque middle layer carrying the mononuclear cells was carefully transferred to a new tube. The mononuclear cells were washed and isolated using magnetic beads and cell sorting. Isolated monocyte purity was assessed and resuspended in Invitrogen™ TRIzol™ reagent. The total RNA extract was stored at −80 °C until use. (2) *Monocyte Differentiation to Macrophage:* For differentiation of monocytes to macrophages, monocytes were enriched by allowing adherence in an incubator. While nonadherent cells were discarded, adherent monocytes were washed. Afterward, the macrophage medium was used for monocyte differentiation into macrophages. M1 macrophages were obtained by treatment with lipopolysaccharides and interferon-γ, while M2 macrophages were obtained by treatment with IL-4.	
B. **Animal**				
Rabbit aortic strips	CAS	1980	Henry et al. [234]	
			**Characteristics:** Both male and female New Zealand White rabbits weighing between 2.0 and 2.5 kg were assigned randomly to 2 dietary groups. One group was maintained on standard pellets, and the other received 2% cholesterol pellets for 9–10 weeks. Then the rabbits were sacrificed and the descending thoracic aorta was quickly excised and cut into strips, then mounted for the measurement of isometric force in an organ bath. Spasm was provoked in atherosclerotic arteries by ergonovine and phenylephrine but not serotonin.	
2. Hamster	Hereditary microvascular CASHFrEF	1982	Factor et al. [220]	Verapamil completely prevented myocardial necrosis and fibrosis and possibly the ultimate development of ventricular failure in the Syrian hamster.
			**Characteristics:** Both male and female Syrian hamsters of the BIO 53.58 strain obtained from Telaco laboratory were evaluated, predominantly at 30, 50, and 150 ± 14 days of age. A few hamsters were also studied at 90 and 210 days of age. Each cardiomyopathic hamster was compared with an age- and sex-matched noncardiomyopathic control shipped in the same batch. Hamsters were fed standard chow. To elucidate the pathogenesis of microvascular CAS, perfusion of silicone rubber solutions revealed numerous areas of microvascular constriction, diffuse vessel narrowing, and luminal irregularity. Pretreatment of young hamsters with verapamil during the period when they developed myocardial necrosis prevented myocytolytic lesions and abolished microvascular hyperreactivity. Hence, focal, transient CAS of small blood vessels, probably secondary to vasoactive substances, may cause myocytolytic necrosis in this model.	
3. Dog	CAS	1982	Noguchi et al. [235]	
			**Characteristics:** Dogs of either sex weighing 14–22 kg were anesthetized with intravenous sodium pentobarbital (25 mg/kg avenously). To obtain maximal vasoconstriction in dogs, 0.4 mg/dog of ergonovine maleate was given intravenously as a bolus. All drugs were diluted with isotonic sodium chloride solution (saline).	
		1984	Kawachi et al. [236]	>50% of coronary narrowing is necessary to induce regional myocardial ischemia during stress in humans and >85% at rest in experimental animals. Ergonovine induced up to only a 40% reduction of coronary luminal diameter, which was too mild to satisfy CAS criteria of >50% of coronary narrowing to cause myocardial ischemia. As a result, dogs are a difficult animal to produce CAS.
			**Characteristics:** In mongrel dogs, selective coronary endothelial denudation by means of cardiac catheterization of either the left anterior or circumflex coronary artery was repeated twice at 1 month intervals. Thereafter, a high-cholesterol diet (20 g/day) was given for 3 and 6 months. No CAS was provoked by intravenous ergonovine before or immediately after endothelial denudation, but a significant reduction in the luminar diameter at the denuded sites, compared with the non-denuded site and the contralateral coronary arteries, was noted angiographically in 1–6 months. A progressive intensity of vasoconstriction in the denuded site after ergonovine was noted for up to 6 months.	
4. Pig	CAS	1983	Shimokawa et al. [237]	
			**Characteristics:** *Endothelium denudation:* Male Gottingen miniature swine (4–6 months of age; 11–22 kg body weight) were fed on a diet containing 2% cholesterol for 3 months after they were subjected to endothelial balloon denudation of the left circumflex coronary artery. CAS was defined as the transient excess vasoconstriction that subsides either spontaneously or after the administration of nitroglycerin and that is characterized by a decrease of >75% in coronary diameter compared with that after the intravenous nitroglycerin (20 μg/kg). CAS was provoked by intracoronary or intravenous histamine in doses of 100 to 400 μg. Ergonovine or serotonin was ineffective to produce CAS. CAS occurred only in the denuded portion of the left circumflex coronary artery.	
		1986	Egashira et al. [238]	The degree of hypercholesterolemia did not affect the provocation of CAS by histamine in pigs.
			**Characteristics:** Thirty-six disease-free male Göttingen miniature pigs that were 4–6 months old and weighed 13–21 kg were fed low-cholesterol regular swine chow before the experiment. The intracoronary histamine-induced CAS before endothelial denudation in 5 of 36 consecutive pigs	
		1996	Shimokawa et al. [239]	
			**Characteristics:** *Endothelium non-denudation:* Male Yorkshire pigs, 2–4 months old and weighing 20–30 kg, were used. They were sedated with intramuscular administration of ketamine hydrochloride (12.5 mg/kg) and anesthetized with intravenous sodium pentobarbital (25 mg/kg). The proximal left anterior and circumflex coronary artery adventitia was treated with IL-1β-bound beads for 1–4 weeks. Intracoronary serotonin, histamine, or platelet-activating factor caused CAS at the IL-1β-treated segment, but not at the control site. Treatment of the adventitia with platelet-derived growth factor also mimicked the effect of IL-1β [240].	
5. Rat	Acquired microvascular CASHFrEF	1985	Sonnenblick et al. [102]	This model will allow specific drug therapy to be designed to prevent the progression of microvascular CASHFrEF.
			**Characteristics:** (1) In this acquired hypertensive diabetic rat model, initial focal myocyte necrosis was followed by focal scars, ventricular hypertrophy, dilatation, HFrEF, and finally death. Microvascular CAS associated with small myocyte necrosis had been shown at an early stage, which then underwent fibrosis. (2) Because (1) the coronary microcirculation is organized as end-capillary loops without connections in the dogs and humans, (2) pretreatment of dogs with phentolamine before embolization prevented myocardial necrosis, and (3) 25–50 μm microspheres being embolized to the coronary microcirculation of dogs and rats leads to focal myocardial necrosis remarkably similar to the cardiomyopathic hamster and the hypertensive-diabetic rat, suggesting that focal myocardial necrosis in hypertensive diabetic rats is caused not only by microembolization but also microvascular CAS.	
	Acquired microvascular CAS	2013	Pearson et al. [241]	
			**Characteristics:** Male Sprague Dawley rats aged 7 weeks old received either a vehicle injection of sodium citrate or streptozotocin to induce type I diabetes. All rats were given food and water ad libitum. Three weeks after vehicle or streptozotocin injection all rats underwent angiography. Endothelium-dependent and -independent response: Serial angiograms were documented at the end of 5 min infusions of vehicle, acetylcholine, and sodium nitroprusside, during vehicle infusion 30 min after inhibited production of both nitric oxide and prostacyclin with Nω-nitro-l-arginine methyl ester and sodium meclofenamate, respectively. A final image series was recorded 10 min after administration of fasudil hydrochloride.	
6. Mice	CAS	1999	Kinjo et al. [242]	
			**Characteristics:** The mouse p122RhoGAP/DLC-1 cDNA was subcloned into a plasmid. The resultant recombinant construct was then microinjected into the pronuclei of fertilized mouse embryos at the single-cell stage to produce transgenic mice (C57BL/6J strain). Then experiments were conducted at the age of 20–30 weeks. After the mice were anesthetized via intraperitoneal drugs injection, then the hearts were quickly excised and transfused via a cannula placed just distal of the intact aortic valve. Coronary arteries were perfused by either ergometrine for 20 min or vehicle, followed by the infusion of Microfil, a liquid latex medium. Coronary angiography with the Microfilms were obtained by X-Ray Inspection Systems	
		2002	Kakkar et al. [129]	No genetic mutation is noted in association with amino acid substitution of SUR in 9 Japanese CAS patients [243].
			**Characteristics:** The 2B isoform of SUR2 (SUR2B) was amplified from a mouse heart cDNA library and placed between the terminal 441 base pairs (bp) of the SM22α promoter and the bovine growth hormone termination and polyadenylation signal sequence. The plasmid (SM22-SUR2B) was injected into fertilized oocytes. Spontaneous CAS and sudden death in SUR2 K_ATP_ null mice arise from a coronary artery vascular smooth muscle-extrinsic process.	
		2003	Chen et al. [130]	Their relevance to CAS in humans remains to be elucidated.
			**Characteristics:** Mice lacking α_1H_ T-type calcium channels have reduced relaxation in response to acetylcholine.	
		2006	Chutkow et al. [128]	No genetic mutation is noted in association with amino acid substitution of SUR in 9 Japanese CAS patients [243].
			**Characteristics:** Episodic CAS and hypertension develop in the absence of SUR2 K_ATP_ channels in SUR2 gene-targeted mice (SUR2^–/–^).	
		2007	Malester et al. [244]	No mutation that alters primary structure of Kir6.1 is detected in 19 Japanese [245] or 18 Italian [246] patients with CAS.
			**Characteristics:** A transgenic mouse model was generatedto specifically target endothelial K_ATP_ channels by expressing a dominant negative Kir6.1 subunit only in the endothelium. There was no evidence of increased susceptibility to ergonovine-induced CAS, but basal endothelin-1 release was significantly elevated in the coronary effluent from these hearts. Spontaneous coronary spasm occurred and consequently led to sudden death.	
		2012	Shibutani et al. [247]	
			**Characteristics:** The R257H variant PLC-δ1 cDNA was subcloned into the plasmid pBsKS(-) including a 4.7-kb fragment of the mouse α-smooth muscle actin promoter. The resultant recombinant construct was microinjected into the pronuclei of fertilized mouse embryos at the single-cell stage to generate transgenic mice (C57BL/6 strain). The experiments were conducted at the age of 20–30 weeks. After anesthesia, ergometrine maleate in two doses (15 and 50 mg/kg) was administered into the mice’s jugular vein over 10 min. The electrocardiogram lead II before and after ergometrine injection was continuously recorded. ST-segment changes, specifically elevation, were in comparison with the baseline electrocardiograms.	
		2013	Yamada et al. [248]	The relevance of SMP30 to CAS in humans remains to be elucidated.
			**Characteristics:** SMP30 knockout mice bred from C57BL/6 mice were generated by a gene-targeting technique. Wild-type C57BL/6 and SMP30 knockout mice (age 8–10 weeks, weight 22.5 ± 2.6 g) were used for the experiments. After the mice were anesthetized with an intraperitoneal drug injection, acetylcholine was administered through a catheter from the cervical artery to the aortic sinus in the SMP30 knockout and wild-type mice. The standard limb leads, aVR, aVL, and aVF were recorded constantly by an electrocardiograph at 1-min intervals.	

## Data Availability

Not applicable.

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
