# Peer review of "Coronary Artery Spasm-Related Heart Failure Syndrome: Literature Review"

_ijms, 2023, doi:10.3390/ijms24087530_

Round 1
Reviewer 1 Report
The review at hand gives a summary of coronary artery spasm-induced heart failure. The review is written with clear and accessible language and presents important points concerning epidemiology, clinical presentation, pathophysiology, and coronary artery spasms-heart failure treatment.
However, I have some comments to consider:
1. Abstract:
The abstract gives a good overview and, overall is intriguing. Though, “Ischemia due to coronary artery disease and coronary artery spasm (CAS) … ”describes two different entities, of which CAD remains by far more important. But I think the abstract could be improved by emphasizing the fact that CAS is undervalued and underdiagnosed, which I believe would make the article more intriguing.
I recommend leaving phenotypes A and B out as they are not explained.
2. Introduction:
While I think the introduction outlines the framework of HFrEF and HFpEF in which CAS-HF is placed very well, the introduction of CAS itself is much too short.
The introduction has to give an outlook on what CAS is defined by.
- Transient myocardial ischemia
- First described 1959 (Prinzmetal)
- Attacks at rest and not provoked by exertion
- Associated with ST-segment elevation
3. Epidemiology:
- Why are Asian patients more likely to be affected by CAS? Why is CAS after intracoronary Acetylcholine so different between Germans and French? Is there any data or hypothesis?
- It also is unclear whether this paragraph contradicts the introductory sentence of this section, stating that HF incidence has decreased in women but not in men.
“In the US, while the incidence and prevalence of HF is increased [11,45],the age-specific incidence of HF might be reduced, but in a less degree in HFpEF compared to HFrEF [46]. A UK study showed that the age-standardized incidence of HF decreased by 7% between 2002 and 2014, whereas the absolute incidence of HF increased by 12%, and prevalent HF increased by 23% [47]. This increase in the absolute number indicates an aging population, improved survival from cardiovascular diseases, including myocardial infarction (MI) [49], and the increasing prevalence of risk factors. Given that approximately 50% of HFrEF cases can be attributed to iscehmia [48], patients with a new diagnosis of HFrEF frequently need an evaluation for underlying CAD. Despite the fact that patient specific factors (eg, advanced age, severe comorbidities, non-candidates for revas-cularization, or no preference to undergo coronary revascularization procedures) should be evaluated prior to referral, coronary angiography is the gold standard for assessment of obstructive CAD [49]. In addition to epicardial CAD, microvascular CAD is becoming widespread and often under-recognized [50]; hence, both epicardial and microvascular CAD, clinically overt or silent, acute or chronic, can result in decreased perfusion, myo-cardial damage and further reduced myocardial function.”
Some of the sources are outdated. Source 32 dates to 2002 and therefore is inappropriate to evaluate trends in heart failure. Since 2002 major breakthroughs in heart failure therapy have been made that this study does not account for.
“Over the past 50 years, the incidence of HF has decreased among females but not among males, whereas survival after the HF onset has improved in both genders [32].”
4. Pathophysiology:
- What are phenotypes A and B? Please explain.
- “Among the risk factors of CAS, smoking (relative risk, 1.47) is independently associated with incident HF [77].” The source provided elucidates that smoking is independently associated with incident HF. Please provide a source demonstrating that smoking is a risk factor for CAS.
- Proposed etiology: While I understand that the mechanisms leading to heart failure in CAS might share similarities with HF due to atherosclerotic coronary artery disease, I think some critical points are neglected:
- Deficient aldehyde dehydrogenase 2 activity
- Smooth muscle cell hyperreactivity could be described in more detail
-etc.
- Many proposed mechanisms appear to be rather hypothetic and extrapolated from other causes of heart failure:
E.g.: “Circulating factors, such as fibrocytes, circulating monocyte-derived cells and fibroblasts, might regulate the effects of microvascular CAS favoring the development of left ventricular fibrosis and hypertrophy”. While I agree that the aforementioned factors might regulate the effects of CAS it’s one of more factors that are speculative.
I would not characterize Tako Tsubo as novel as it was discovered in the 90s.
5. Treatment:
While some treatment options are mentioned both in the graphic and text, garcinol is not mentioned in the text. Garcinol should be discussed in the treatment section.
6. Convoluted sentences:
Although most of the review is very well written and easy to understand, some sentences appear convoluted and could be shortened. E.g.: “Taken together, although no guideline addresses the therapeutic significance of calcium channel blockers in CASHF [24], the differential diagnosis of dilated cardiomyopathy or HFrEF should include CAS since calcium channel blockers are potentially promising medical options [24,75], and after HF condition is stabilized, the provocative testing for CAS can be safely performed [25].”
7. Conclusion
I disagree with your statement that “CASHF should be regarded as medical failure rather than indication for starting treatment.” As I understand, underappreciation of CAS-mediated HF is a medical failure that occurs very often. However, it does not change the necessity to start treatment because as soon as you recognize CAS as the culprit, you would treat CAS as well as HF. “Nonetheless, while HF treatment aims to control the symptoms and slow down the progression, CASHF is one of a few conditions that medical therapy may reverse HF.”
8. Figure 1:
While the translation of molecular and cellular mechanisms to clinical findings and treatment is depicted nicely, the dimension of the figure could be improved upon. Especially the cellular and molecular mechanisms in the figure’s upper part should be enlarged to display a similar font size as the lower part. It would make the figure easier to read and give more importance to the pathophysiologic mechanisms.
Overall, the review at hand elucidates an intriguing and important, yet in clinical practice often neglected aspect of heart failure. Increased awareness for this complex entity is certainly necessary to move clinical practice forward. At this point though, the knowledge concerning CAS-HF appears to be very limited and in large parts to be deduced from observations from other models and pathologic entities. Nevertheless, if the topic at hand would be described more focused, the review at hand would provide very interesting points and an important stimulus to appreciate the role of CAS-induced HF.
Author Response
Reviewer 1#
The review at hand gives a summary of coronary artery spasm-induced heart failure. The review is written with clear and accessible language and presents important points concerning epidemiology, clinical presentation, pathophysiology, and coronary artery spasms-heart failure treatment.
However, I have some comments to consider:
- Abstract: The abstract gives a good overview and, overall is intriguing. Though, “Ischemia due to coronary artery disease and coronary artery spasm (CAS) ”describes two different entities, of which CAD remains by far more important. But I think the abstract could be improved by emphasizing the fact that CAS is undervalued and underdiagnosed, which I believe would make the article more intriguing. I recommend leaving phenotypes A and B out as they are not explained.
Response to reviewer:
We appreciate the reviewer’s comments. While coronary artery spasm (CAS) is still underappreciated and may be misdiagnosed, ischemia due to coronary artery disease and CAS is becoming the single most frequent cause of HF worldwide. We have deleted phenotypes A and B in the abstract of this revised manuscript (Page 1). Thank you for your comment.
- Introduction: While I think the introduction outlines the framework of HFrEF and HFpEF in which CAS-HF is placed very well, the introduction of CAS itself is much too short. The introduction has to give an outlook on what CAS is defined by: Transient myocardial ischemia, First described 1959 (Prinzmetal), Attacks at rest and not provoked by exertion, Associated with ST-segment elevation.
Response to reviewer:
We appreciate the reviewer’s comments. Dr. Myron Prinzmetal (1908–1987) published his observations on "A variant form of angina pectoris" in 1959, which was the 1st article [26] distinguishing it as a distinct entity from the classic angina pectoris (pectoris dolor) described by Dr. William Heberden (1710–1801) based on 20 cases with this affliction in 1772 [27], which occurred when increased cardiac work or emotional disturbance provoked chest pain and was relieved by rest or the administration of nitroglycerin. In Prinzmetal’s 1st report of 32 cases of variant angina, of which 20 were personally observed and 12 were reported in literature between 1931-1956, the pain associated with transient non-progressive ST-segment elevation appeared at rest or during ordinary activity but was not brought on by exercise or emotional disturbance. Among the 32 patients studied, 12 developed myocardial infarction during follow-up [26]. Because coronary atherosclerosis was a common finding in both forms of angina pectoris post mortem, and the attack usually occurred with the subject at rest, when vascular hypertonic activity is physiologically greatest [26], vascular hypertonus proposed by Prinzmetal et al. or CAS proposed by other researchers was the explanation for variant angina. Although CAS had never been proved [28,29] within a decade following the 1st report of coronary angiography in 1959, CAS was documented angiographically in early 1970s in patients of variant angina [29,30,31]. In the 1970s and 1980s, the diagnosis of CAS by coronary angiography in the catheterization laboratory was not rare. It then became clear that CAS could occur in patient with atherosclerotic obstructive CAD [26,27] or angiographically normal coronary arteries, which was referred to as “variant of the variant” [30]. Moreover, ST-segment depression rather than non-progressive elevation occurred more commonly in CAS [32,33]. Therefore, the term “variant angina” is used for CAS-related angina with transient non-progressive ST-segment elevation (Figure 1). Additionally, CAS and acute coronary events can be caused by allergic reactions, with mediators released during mast cell degranulation such as histamine, chymase, leukotrienes, platelet activating factor acting on coronary vascular smooth muscle cells that constitute the pathophysiologic basis of Kounis syndrome [34,35]. Collectively, the presence of atherosclerotic obstructive coronary artery disease cannot be considered as the only determinant of angina pectoris [36]. Most importantly, the use of nitroglycerin at the beginning of coronary angiography should be avoided [37]. However, the nitroglycerin solution must be fully prepared before performing CAS provocative testing to abolish established CAS promptly through intracoronary administration [33]. Therefore, 2 sets of coronary angiograms before and after intracoronary nitroglycerin should be obtained routinely once obstructive lesions are noted. Spontaneous CAS can be misdiagnosed as a candidate for percutaneous coronary intervention unless the relief of obstructive stenosis is documented after intracoronary nitroglycerin administration, emphasizing the importance of intracoronary nitroglycerin administration before attempted coronary intervention, and avoiding unnecessary coronary revascularization [38]. We have added the above statements to the “Introduction” section in this revised manuscript (Page 2,3,4). Thank you for your comment.
- Epidemiology:
- Why are Asian patients more likely to be affected by CAS? Why is CAS after intracoronary Acetylcholine so different between Germans and French? Is there any data or hypothesis?
- It also is unclear whether this paragraph contradicts the introductory sentence of this section, stating that HF incidence has decreased in women but not in men.
“In the US, while the incidence and prevalence of HF is increased [11,45],the age-specific incidence of HF might be reduced, but in a less degree in HFpEF compared to HFrEF [46]. A UK study showed that the age-standardized incidence of HF decreased by 7% between 2002 and 2014, whereas the absolute incidence of HF increased by 12%, and prevalent HF increased by 23% [47]. This increase in the absolute number indicates an aging population, improved survival from cardiovascular diseases, including myocardial infarction (MI) [49], and the increasing prevalence of risk factors. Given that approximately 50% of HFrEF cases can be attributed to iscehmia [48], patients with a new diagnosis of HFrEF frequently need an evaluation for underlying CAD. Despite the fact that patient specific factors (eg, advanced age, severe comorbidities, non-candidates for revas-cularization, or no preference to undergo coronary revascularization procedures) should be evaluated prior to referral, coronary angiography is the gold standard for assessment of obstructive CAD [49]. In addition to epicardial CAD, microvascular CAD is becoming widespread and often under-recognized [50]; hence, both epicardial and microvascular CAD, clinically overt or silent, acute or chronic, can result in decreased perfusion, myo-cardial damage and further reduced myocardial function.”
Some of the sources are outdated. Source 32 dates to 2002 and therefore is inappropriate to evaluate trends in heart failure. Since 2002 major breakthroughs in heart failure therapy have been made that this study does not account for.
“Over the past 50 years, the incidence of HF has decreased among females but not among males, whereas survival after the HF onset has improved in both genders [32].”
Response to reviewer:
We appreciate the reviewer’s comments. Despite the fact that the cumulative incidence of HF is similar between both genders, women are approximately 65% less likely to develop HFrEF than men, particularly in their younger years [45-47], while HFpEF is twice as common in women than men, which results from physiologic differences between the 2 genders [48]. On the other hand, survival after a diagnosis of HF, irrespective of HFrEF or HFpEF, has shown modest improvement in the 21st century and lags behind other serious conditions, such as cancer [49]. CAS is an exceptionally complex multifactorial disease in which smoking, inflammation, metabolic, psychosocial and physical factors come into play. Although it was reported more than 20 years ago [50], the racial differences of coronary vasomotion disorders between Asian and Caucasian populations remain controversial. First, previous studies show that epicardial CAS is more often recognized in Japanese and Taiwanese people than in Caucasian populations, while CMD is typically observed in Caucasian patients, which may be because Japanese and Taiwanese cardiologists have perform spasm provocation testing actively for 30 and 20 years, respectively, in patients with unobstructive CAD, whereas most Caucasian cardiologists do not perform provocative testing for nonobstructive CAD in the cardiac catheterization laboratory [59]. Second, various diagnostic procedures are performed worldwide, such as intravenous ergonovine-provoked >70% luminal reduction in France [60] and intracoronary acetylcholine-provoked >75% luminal reduction in Germany [56]. Third, according to a Japanese study [61], intracoronary acetylcholine administration time is crucial to provoke CAS. Slow injection of acetylcholine for 3 minutes may induce microvascular CAS, whereas rapid injection of acetylcholine for 20-30 seconds may provoke epicardial CAS, leading to inconsistency in the prevalence and incidence of CAS between Japanese and Caucasian patients. Fourth, the Japanese cardiologist have stated that in some European institutions, acetylcholine testing without pacemakers is employed, which may cause bradycardia or cardiac arrest in the right coronary artery rendering difficult interpretation of provocative testing. If Caucasian cardiologists perform provocative testing with pacemakers similar to Japanese cardiologists, the prevalence and incidence of CAS may be higher than ever thought. However, in Taiwanese specialists’ experience without pacemakers when performing provocative testing using the bolus injection of ergonovine, there has been no cardiac arrest but only rarely mild bradycardia, which can quickly return to normal after immediate intracoronary administration of nitroglycerin once CAS occurs [62,63]. Fifth, the definition of positive epicardial CAS is different among previous Japanese, Taiwanese and Caucasian studies [59]. For example, provoked CAS is defined as a reduction of >50% [64], >70% [60,65,66], >75% [56,67-70], >90% [24,51,70,71], or 99-100% [72] in luminal diameter compared with postintracoronary nitroglycerin. Sixth, among all the cluster of CAS risk factors, the predominant factors that cause CAS in Asian patients may be different from those in white patients. Notably, it is estimated that nearly 1 billion people globally, most of whom are Asians, carry the Glu504Lys polymorphism in the aldehyde dehydrogenase 2 (ALDH2) gene [73]. This ALDH2 mutant is significantly associated with high level of high-sensitivity C-reactive protein [74], which is a risk factor for CAS. We have added the above statements to the “Epidemiology” section in this revised manuscript (Page 4,5). Thank you for your comment.
- Pathophysiology:
- What are phenotypes A and B? Please explain.
- “Among the risk factors of CAS, smoking (relative risk, 1.47) is independently associated with incident HF [77].” The source provided elucidates that smoking is independently associated with incident HF. Please provide a source demonstrating that smoking is a risk factor for CAS.
- -Proposed etiology: While I understand that the mechanisms leading to heart failure in CAS might share similarities with HF due to atherosclerotic coronary artery disease, I think some critical points are neglected:
- Deficient aldehyde dehydrogenase 2 activity
- Smooth muscle cell hyperreactivity could be described in more detail
-etc.
- Many proposed mechanisms appear to be rather hypothetic and extrapolated from other causes of heart failure: E.g.: “Circulating factors, such as fibrocytes, circulating monocyte-derived cells and fibroblasts, might regulate the effects of microvascular CAS favoring the development of left ventricular fibrosis and hypertrophy”. While I agree that the aforementioned factors might regulate the effects of CAS it’s one of more factors that are speculative.
- I would not characterize Tako Tsubo as novel as it was discovered in the 90s.
Response to reviewer:
We appreciate the reviewer’s comments. CAS not only cause remarkable progression of CAD but also demonstrates a “jump-up” phenomenon, as short as in 25 minutes in a swine model [115], in which the development of CAS superimposed on a minimal coronary stenosis can progresses to total atherosclerotic obstruction, resulting in myocardial infarction [116]. In a swine model, the phenomenon can be explained by that CAS of abrupt, but not of gradual onset, can cause intramural hemorrhage in the plaque’s neovasculature and the subsequent sudden progression of organic coronary stenosis, leading to myocardial infarction [117]. Most importantly, while smoking, age, C-reactive protein (CRP) [50], ALDH2 deficiency [118] and lipoprotein(a) [119] are risk factors for CAS, CAS is not associated with the traditional risk factors for CAD [50,120], such as diabetes mellitus, hypertension [53], hypercholesterolemia [50,120] and obesity [50,120], suggesting pathophysiological differences exist between CAS and CAD. While risk factors in an individual often coexist and interact with one another (Figure 3), precipitating factors refer to a specific event. Notably, while the older people are more likely to develop CAS than their younger counterparts, smoking in the younger compared to their older analogs has a more powerful effect on CAS occurrence [121]. In addition, these factors may be gender-specific, as smoking and age appear to have a more significant role in men [122]. Vascular smooth muscle cell relaxation and contraction are regulated mainly through myosin light chain dephosphorylation and phosphorylation, respectively. CMD is characterized by impaired microvascular smooth muscle cell dilation, which can ultimately lead to HFpEF [123]. In addition, increased smooth muscle cell Rho-kinase activity favors contraction by directly increasing sensitization to Ca2+ of myosin light chain and indirectly augmenting myosin light chain phosphorylation [124]. Shimokawa et al. showed that Rho-kinase activity is enhanced in coronary artery smooth muscle cell after wrapping the coronary artery with interleukin-1β beads in a porcine model of CAS [124,125,126]. Other experimental models of spontaneous CAS have been developed in KATP mutant or SUR2 KATP knockout mice, suggesting that loss of function of KATP channels causes smooth muscle cell hypercontraction in the absence of atherosclerotic lesions [127,128]. Mice deficient in α1H T-type calcium channels demonstrate normal coronary artery contraction, but reduced relaxation in response to acetylcholine [129]. Together, these models show that vascular smooth muscle cell hyperreactivity can cause CAS through different pathways; however, their relevance to CAS in humans remains to be elucidated. Because the majority of CASHF patients presents with the less severe phenotype (stages A and B) than overt HF, among which stage A is for patients at risk for HF but without current or prior symptoms or signs of HF and without structural or biomarker evidence of heart disease, and stage B is for patients without current or prior symptoms or signs of HF but evidence of structural heart disease or abnormal cardiac function, or elevated natriuretic peptide levels [130], it underscores the importance of understanding risk factors for CAS to prevent the future burden of HF. As researchers associated with the National Human Genome Research Institute unlock the mysteries of the human genome (the complete set of human genes), almost all diseases have a genetic component [148], which is the case with CAS in that a mutation in the ALDH2 gene is believed to be the cause [149] and associated with Asian flush syndrome. Mizuno Y et al. has shown that Asians with defective ALDH2*2 alleles have a higher risk of CAS. They also found that the defective gene positively interacts with the detrimental effects of smoking on stronger vasoconstriction than each factor alone by increasing reactive aldehydes [150]. Furthermore, other Japanese studies show that the mutant ALDH2*2 allele carriers compared to subjects with ALDH2*1/1 genotype have higher frequencies of more severe CAS-related myocardial injury [118,151]. Takotsubo cardiomyopathy is an acute and reversible form of HFrEF featuring symptoms and signs of acute MI without CAD, in which the apex of the left ventricle balloon and enlarge to resemble a takotsubo, a Japanese octopus pot [180]. We have added the above statements to the “Pathogenesis” section in this revised manuscript (Page 7,8,9,13). Thank you for your comment.
- Treatment: While some treatment options are mentioned both in the graphic and text, garcinol is not mentioned in the text. Garcinol should be discussed in the treatment section.
Response to reviewer:
We appreciate the reviewer’s comments. In acute myocardial infarction, although timely coronary reperfusion is the most effective way to limit myocardial injury, the subsequent cardiomyocyte apoptosis, adverse left ventricular remodeling and, finally, ischemic HF, the medical therapies of the ensuing tissue inflammation and its subsequent active suppression and resolution, remains elusive [196]. Among natural dietary plant-derived phytochemicals that have been investigated to protect from a number of chronic diseases, garcinol revealed its potential therapeutic effects in in vitro studies, such as its anti-oxidative, anti-inflammatory, and anti-cancer properties [197]. In rat models of with isoproterenol-induced HFrEF, garcinol treatment increased the heart rate and improved the maximum rate of increase in pressure (+dp/dtmax), maximum rate of decrease in pressure (–dp/dtmax), ejection fraction and systolic pressure in the left ventricle [198]. We have previously demonstrated that garcinol suppresses lipoprotein(a)-induced oxidative stress and inflammatory cytokines by α7-nicotinic acetylcholine receptor-mediated inhibition of p38 MAPK/NF-κB signaling in cardiomyocyte AC16 cells and isoproterenol-induced acute myocardial infarction mice [199]. These observations suggest that garcinol may effectively prevent cardiomyocyte apoptosis. We have added the above statements to the “Treatment” section in this revised manuscript (Page 20). Thank you for your comment.
- Convoluted sentences: Although most of the review is very well written and easy to understand, some sentences appear convoluted and could be shortened. E.g.: “Taken together, although no guideline addresses the therapeutic significance of calcium channel blockers in CASHF [24], the differential diagnosis of dilated cardiomyopathy or HFrEF should include CAS since calcium channel blockers are potentially promising medical options [24,75], and after HF condition is stabilized, the provocative testing for CAS can be safely performed [25].”
Response to reviewer:
We appreciate the reviewer’s comments. We have deleted the sentence “and after HF condition is stabilized, the provocative testing for CAS can be safely performed [25].” in the “Treatment” section in this revised manuscript (Page 20). Thank you for your comment.
- Conclusion: I disagree with your statement that “CASHF should be regarded as medical failure rather than indication for starting treatment.” As I understand, underappreciation of CAS-mediated HF is a medical failure that occurs very often. However, it does not change the necessity to start treatment because as soon as you recognize CAS as the culprit, you would treat CAS as well as HF. “Nonetheless, while HF treatment aims to control the symptoms and slow down the progression, CASHF is one of a few conditions that medical therapy may reverse HF.”
Response to reviewer:
We appreciate the reviewer’s comments. Because CAS can cause rapid plaque progression of CAD and the development of acute coronary syndrome, including myocardial infarction, underrecognized CAS is concerned with the health of the individuals and population as a whole, as well as with the health implications of the economic and social policies, and investment in health policies. While medical care can prolong survival and improve prognosis after occurrence of CAD and HF, more important is to identify ill people afflicted with CAS before the potential subsequent development of CAD and HF in the first place. It has been demonstrated that after HF condition is stabilized, the provocative testing for CAS can be safely performed. Treatment should be started early once CAS is diagnosed. On the other hand, while HF treatment aims to control the symptoms and slow down the progression, CASHF is one of a few conditions that medical therapy may reverse HF. We have reworded and added the above statements to the “Conclusions” section in this revised manuscript (Page 21). Thank you for your comment.
- Figure 1: While the translation of molecular and cellular mechanisms to clinical findings and treatment is depicted nicely, the dimension of the figure could be improved upon. Especially the cellular and molecular mechanisms in the figure’s upper part should be enlarged to display a similar font size as the lower part. It would make the figure easier to read and give more importance to the pathophysiologic mechanisms.
Response to reviewer:
We appreciate the reviewer’s comments. We have modified Figure 4 as the reviewer suggested (Page 22). Thank you for your comment.

Reviewer 2 Report
Ming-Jui Hung et al. present in a review article the role of CAS and the association to heart failure. The article is well writen. However, some points should be improved.
1-Please add an illustrative picture of pathomechanisms of CAS.
2-You need to add an models, which tested the pathomechanism of CAS, animal models?
3-For a long time it was discussed that takotsubo syndrome is also related to CAS. Please discuss this point.
4-What is the association between CAS and arrhythmias?
5-You need some illustrative figures of coronary vessels showing before and after vasospasm.
What is the role of Nitro wihtin a a coronary angiography to confirm CAS?
6-Are they ECG changes for CAS? e.g. ST-segment elevation?
Author Response
Reviewer 2#
Ming-Jui Hung et al. present in a review article the role of CAS and the association to heart failure. The article is well written. However, some points should be improved.
- Please add an illustrative picture of pathomechanisms of CAS.
Response to reviewer:
We appreciate the reviewer’s comments. We have added Figure 3 as an illustrative picture of pathomechanisms of CAS (Page 8). Thank you for your comment.
- You need to add an models, which tested the pathomechanism of CAS, animal models?
Response to reviewer:
We appreciate the reviewer’s comments. We have added “4.3 Cellular and Animal Models of Takotsubo Cardiomyopathy, CAS and microvascular CASHrEF” and “Table 2. Proposed models of Takotsubo cardiomyopathy and coronary artery spasm.” to the “Pathogenesis” section in this revised manuscript (Page 14-19). Thank you for your comment.
- For a long time it was discussed that takotsubo syndrome is also related to CAS. Please discuss this point.
Response to reviewer:
We appreciate the reviewer’s comments. Takotsubo cardiomyopathy is an acute and reversible form of HFrEF featuring symptoms and signs of acute MI without CAD, in which the apex of the left ventricle balloon and enlarge to resemble a takotsubo, a Japanese octopus pot [180]. The syndrome is precipitated by abrupt, unexpected emotional distress and more frequently occurs in older women than in men [180]. Precipitating mechanisms are multifactorial and complex, including microvascular and epicardial CAS [180], genetics and thyroid disorders [181]. Stress activates the sympathetic nervous system to release circulatory catecholamines and the hypothalamic-pituitary-adrenal axis to release circulatory glucocorticoids [182]. While initially protective for the heart, glucocorticoids not only increase plasma levels of catecholamines by inhibiting uptake but also induce cardiac supersensitivity to catecholamines, leading to enhanced β-adrenoceptor signal transduction system [182]. Excessive catecholamines can cause diminished ventricular apical wall motion and enhanced basal motion due to the apicobasal adrenoceptor gradient [181]. Furthermore, low catecholamine levels stimulate cardiac Ca2+ movements, whereas excessive catecholamine levels induce intracellular Ca2+ overload in cardiomyocytes, resulting in cardiac dysfunction [183]. On the other hand, under stressful conditions, high catecholamine levels are oxidized to form oxyradicals, which can cause CAS [182]. Few cases of Takotsubo cardiomyopathy due to angiographically confirmed focal, single vessel, or multivessel CAS have been reported. A retrospective analysis in 10 of 48 (21%) Takotsubo cardiomyopathy cases have shown positive provocative CAS, 5 of whom involved both right and left coronary arteries [184]. Angelini reported 4 cases of Takotsubo cardiomyopathy in which echocardiographic apical ballooning or similar symptoms could be reproduced by provocative CAS [185]. Moreover, it has also been demonstrated that alternate recurrent CAS and Takotsubo cardiomyopathy can exist in the same individual [186]. These observations underscore the importance of CAS as a culprit process underlying Takotsubo cardiomyopathy and the targeted treatments accordingly. Further studies will provide critical insights into this unique issue. We have added the above statement to the “4.2. Epicardial CASHF” section in this revised manuscript (Page 13). Thank you for your comment.
- What is the association between CAS and arrhythmias?
Response to reviewer:
We appreciate the reviewer’s comments. Arrhythmias, particularly ventricular, appear more frequently through unknown mechanisms during CAS attacks in >50% of cases than during attacks of classic Heberden's angina pectoris [39,107]. Ventricular arrhythmias are more common during anterior wall ischemia [108]. Sudden death with normal appearing coronary arteries on autopsy examination has been attributed to VF complicating CAS [109]. Although VF uisually needs to be terminated by cardioversion, CAS-related VF rarely reverts spontaneously [50,80] (Figure 2). In addition, VF was found to be asymptomatic in 43% and nonsustained in 40% episodes in a study of patients with implantable cardioverter defibrillators [110]. The incidence of syncope or pre-syncope is 25% when VF is <10 seconds, compared with 62% if VF is ≥10 seconds [110]. On the other hand, about 40% of CAS-related VF and inferolateral J wave does not cause angina at the first VF, and could have been misdiagnosed as early repolarization syndrome [111], which in the younger age group is associated with features of CAS such as lower systolic blood pressure and lower heart rate [112]. Therefore, CAS is essential and should be considered in the differential diagnosis of syncope and early repolarization syndrome, prompting optimal medical management. On the other hand, severe CAS may cause fatal pulseless electrical activity or asystole without complications of ventricular tachycardia or VF [50]. Triple-vessel severe CAS can cause the heart to suddenly stop beating due to pulseless electrical activity and flash-freeze the entire myocardium in an instant, resulting in unrecognized coronary flow [113]. Consequently, contrast medium may stay in the coronary arteries for a prolonged period of time despite intracoronary administration of nitroglycerine. Prolonged continuous cardiac massage has been effective for resolving CAS-related pulseless electrical activity [113]. However, cardiac pacing or implantable cardioverter defibrillator might not restore frozen myocardium to viable muscle during pulseless electrical activity arrest, and may lead to unexplained death after the implantation [113,114]. Furthermore, CAS-related ischemia of the sinus node artery or atrioventricular node artery can influence the occurrence of pulseless electrical activity or asystole [113]. Collectively, CAS may cause pulseless electrical activity or asystole without ventricular arrhythmias. We have added the above statement, including Figure 2, to the “3. Clinical features of CASHF” section in this revised manuscript (Page 6-7). Thank you for your comment.
- You need some illustrative figures of coronary vessels showing before and after vasospasm.
Response to reviewer:
We appreciate the reviewer’s comments. We have added Figure 1 to the “Introduction” section in this revised manuscript (Page 3). Thank you for your comment.
- What is the role of Nitro wihtin a coronary angiography to confirm CAS?
Response to reviewer:
We appreciate the reviewer’s comments. Most importantly, the use of nitroglycerin at the beginning of coronary angiography should be avoided [37] to prevent inadvertent abrogation of spontaneous CAS. However, the nitroglycerin solution must be fully prepared before performing CAS provocative testing to abolish established CAS promptly through intracoronary administration [33]. Therefore, 2 sets of coronary angiograms before and after intracoronary nitroglycerin should be obtained routinely once obstructive lesions are noted. Spontaneous CAS can be misdiagnosed as a candidate for percutaneous coronary intervention unless the relief of obstructive stenosis is documented after intracoronary nitroglycerin administration, emphasizing the importance of intracoronary nitroglycerin administration before attempted coronary intervention, and avoiding unnecessary coronary revascularization [38]. We have added the above statement to the “Introduction” section in this revised manuscript (Page 3-4). Thank you for your comment.
- Are they ECG changes for CAS? e.g. ST-segment elevation?
Response to reviewer:
We appreciate the reviewer’s comments. We have added Figure 1 illustrating ST-segment elevation during CAS to the “Introduction” section in this revised manuscript (Page 3). Thank you for your comment.
Reviewer 3 Report
The review focuses on the coronary spasm induced microcirculatory dysfunction
Although the review shows some interesting aspects it lacks of focus and the length is rather extensive.
The term INOCA is not clearly defined throughout the manuscript and the comments on Takotsubo seem out of reason.
Generally, the paper should be reduced by 30% and focus only on the pathophysiology of spasm.
Author Response
Reviewer 3#
The review focuses on the coronary spasm induced microcirculatory dysfunction. Although the review shows some interesting aspects it lacks of focus and the length is rather extensive.
- The term INOCA is not clearly defined throughout the manuscript and the comments on Takotsubo seem out of reason.
Response to reviewer:
We appreciate the reviewer’s comments. INOCA is defined as when patients present with symptoms and signs suggesting ischemia but found to have no obstructive CAD at coronary angiography [83]. Takotsubo cardiomyopathy is an acute and reversible form of HFrEF featuring symptoms and signs of acute MI without CAD, in which the apex of the left ventricle balloon and enlarge to resemble a takotsubo, a Japanese octopus pot [180]. The syndrome is precipitated by abrupt, unexpected emotional distress and more frequently occurs in older women than in men [180]. Precipitating mechanisms are multifactorial and complex, including microvascular and epicardial CAS [180], genetics and thyroid disorders [181]. Stress activates the sympathetic nervous system to release circulatory catecholamines and the hypothalamic-pituitary-adrenal axis to release circulatory glucocorticoids [182]. While initially protective for the heart, glucocorticoids not only increase plasma levels of catecholamines by inhibiting uptake but also induce cardiac supersensitivity to catecholamines, leading to enhanced β-adrenoceptor signal transduction system [182]. Excessive catecholamines can cause diminished ventricular apical wall motion and enhanced basal motion due to the apicobasal adrenoceptor gradient [181]. Furthermore, low catecholamine levels stimulate cardiac Ca2+ movements, whereas excessive catecholamine levels induce intracellular Ca2+ overload in cardiomyocytes, resulting in cardiac dysfunction [183]. On the other hand, under stressful conditions, high catecholamine levels are oxidized to form oxyradicals, which can cause CAS [182]. Few cases of Takotsubo cardiomyopathy due to angiographically confirmed focal, single vessel, or multivessel CAS have been reported. A retrospective analysis in 10 of 48 (21%) Takotsubo cardiomyopathy cases have shown positive provocative CAS, 5 of whom involved both right and left coronary arteries [184]. Angelini reported 4 cases of Takotsubo cardiomyopathy in which echocardiographic apical ballooning or similar symptoms could be reproduced by provocative CAS [185]. Moreover, it has also been demonstrated that alternate recurrent CAS and Takotsubo cardiomyopathy can exist in the same individual [186]. These observations underscore the importance of CAS as a culprit process underlying Takotsubo cardiomyopathy and the targeted treatments accordingly. Further studies will provide critical insights into this unique issue. We have added the definition of INOCA to the “3. Clinical features of CASHF” section in this revised manuscript (Page 5) and the “4.2. Epicardial CASHF” section in this revised manuscript (Page 13). Thank you for your comment.
- Generally, the paper should be reduced by 30% and focus only on the pathophysiology of spasm.
Response to reviewer:
We appreciate the reviewer’s comments. Because 15 questions were raised by the Reviewers 1# and 2#, it may not be possible to reduce the context by 30% at this time. However, I agree with the reviewer’s concern, and will discuss with the editors if some of the context can be moved to “Supplementary Materials”. Thank you for your comment.
Round 2
Reviewer 2 Report
Thank you for revising the paper
Author Response
Dear reviewer 2,
Thank you for your comments.
Thank you for giving me the chance to revise the paper.